# Spared perilesional V1 activity underlies training-induced recovery of luminance detection sensitivity in cortically-blind patients

Antoine Barbot[1,2,3], Anasuya Das[1], Michael D. Melnick[2,4], Matthew R. Cavanaugh[1,2], Elisha P. Merriam[5,6,8], David J. Heeger [5,7,8] & Krystel R. Huxlin [1,2,4,8 ✉]

Damage to the primary visual cortex (V1) causes homonymous visual-field loss long considered intractable. Multiple studies now show that perceptual training can restore visual functions in chronic cortically-induced blindness (CB). A popular hypothesis is that training can harness residual visual functions by recruiting intact extrageniculostriate pathways. Training may also induce plastic changes within spared regions of the damaged V1. Here, we link changes in luminance detection sensitivity with retinotopic fMRI activity before and after visual discrimination training in eleven patients with chronic, stroke-induced CB. We show that spared V1 activity representing perimetrically-blind locations prior to training predicts the amount of training-induced recovery of luminance detection sensitivity. Additionally, training results in an enlargement of population receptive fields in perilesional V1, which increases blind-field coverage and may support further recovery with subsequent training. These findings uncover fundamental changes in perilesional V1 cortex underlying training-induced restoration of conscious luminance detection sensitivity in CB.

[1] Flaum Eye Institute, University of Rochester Medical Center, Rochester, NY, USA. [2] Center for Visual Science, University of Rochester, Rochester, NY, USA. [3] Spinoza Centre for Neuroimaging, Amsterdam, The Netherlands. [4] Brain and Cognitive Sciences, University of Rochester, Rochester, NY, USA. [5] Center for Neural Science, New York University, New York, NY, USA. [6] National Institute of Mental Health, NIMH/NIH, Bethesda, MD, USA. [7] Department of Psychology, New York University, New York, NY, USA. [8]These authors contributed equally: Elisha P. Merriam, David J. Heeger, Krystel R. Huxlin. ✉email: khuxlin@ur.rochester.edu

The primary visual cortex (V1) is the chief cortical relay of visual information from retino-geniculate centers towards higher-level extrastriate areas. Unilateral damage to V1 or its immediate afferents (optic radiation) causes cortically-induced blindness (CB)–a homonymous loss of conscious vision over the contralateral hemifield. Strokes (either ischemic or hemorrhagic) involving the posterior or middle cerebral arteries account for the majority of cases[1,2]. Although CB patients often show damage to extrastriate visual areas, it is damage to V1 that primarily causes the loss of conscious vision and the appearance of defects in luminance contrast detection[3,4], routinely measured using clinical perimetry devices. The prevalence of blindness-inducing post-chiasmal lesions of the visual system in the general population is remarkably high[5], and its impact on everyday life deeply debilitating[6]. Yet, there is a lack of validated clinical therapies that can help restore, rather than compensate for, deficits in CB patients[4,5,7–10]. In stark contrast with well-established physical therapies prescribed following motor cortex strokes, the ability of different restitution therapies to restore vision in CB patients remains highly variable[4,7,11,12].

Spontaneous recovery typically occurs during the first 3 months post-lesion, with little improvement observed following this sub-acute phase, and significant changes no longer observed in chronic patients, >6 months post-lesion[1,2]. The first major commercial restoration treatment for chronic CB patients–NovaVision's Visual Restoration Therapy (VRT)[13]–proposed that intensive computer-based training on a microperimetric luminance detection task could help restore visual sensitivity and shrink perimetrically-defined blind fields in CB patients. Although early findings generated strong interest, subsequent work identified flaws in NovaVision's approach, which when corrected, revealed it to be relatively ineffective[14,15]. Despite this initial failure, which reinforced the clinical dogma that post-stroke vision restoration is not possible, scientific teams worldwide subsequently showed that intensive training with gaze-contingent stimulus presentation inside the blind field can restore a range of visual functions at trained, blind-field locations[4,8,9,16–27]. Critically, gaze-contingent stimulus presentation using binocular eyetracking during both pre- and post-training tests ensured that improvements within the blind field could not simply be explained by the development of compensatory eye movement strategies. Rehabilitation studies in chronic CB have in common the fact that, unlike perceptual learning with intact vision[18,28], significant improvements at blind-field locations require daily, retinotopically-specific training over weeks to months. This and other differences between training in intact *versus* damaged brains remain unexplained, partly due to our poor understanding of the capacity of damaged, adult visual systems for perceptual processing, plasticity and ultimately learning. Given that we now possess the means to reliably induce recovery in CB, we are in an ideal position to assess the neural mechanisms underlying this phenomenon in V1-damaged individuals.

One proposed recovery mechanism is that training stimulates and improves visual processing in extrageniculostriate pathways mediating blindsight[7–9,11,29,30]. Both human and non-human primates with V1 lesions exhibit visually-guided perceptual abilities within their blind field, despite lacking awareness[3,30–33]. Because blindsight is elicited by large stimuli with high-temporal and low-spatial frequency content, it is thought to rely mainly on direct geniculo-hMT+ and/or superior colliculus-pulvinar-extrastriate projections[3,7,30,31,34–36]. Accordingly, some rehabilitation approaches have targeted blindsight to help CB patients recruit these unconscious processes, with some evidence of increased visual performance and awareness post-training[4,8,9,16,27,37]. However, training can also recover the ability to discriminate visual information with spatio-temporal properties and motion integration requirements that fail to elicit blindsight[4,11,20–25].

Thus, intact extrageniculostriate pathways mediating blindsight may not be the only means by which training can elicit recovery in CB. This is particularly encouraging given the restricted bandwidth of blindsight abilities and the fact that not all CB patients exhibit blindsight[8,24].

Another, not mutually-exclusive possibility is that visual recovery in CB relies on spared, perilesional V1 cortex that is functionally impaired, but can be brought back "online" using visual training[11,25,29,38]. Here, we directly tested the potential role of spared V1 cortex in training-induced recovery of luminance detection sensitivity in eleven adult patients with chronic, stroke-induced CB. To do so, we compared retinotopic fMRI activity in early visual cortex with changes in luminance detection sensitivity induced by visual discrimination training, but measured with Humphrey Visual Field (HVF) perimetry. By relying on stringent inclusion criteria and a more uniform patient group than most prior studies, we identified ubiquitous, functional changes mediating recovery in chronic, stroke-induced CB. We show that training-induced recovery of conscious luminance detection sensitivity in chronic CB relies upon spared, visually-evoked activity in perilesional V1 cortex representing regions of the blind field prior to training. Furthermore, changes in perilesional V1 activity following training increased visual coverage of the blind field, which may support further perceptual recovery with subsequent training. Limited changes were observed in extrastriate areas (V2-V4). As such, our results provide vital insights regarding potentially ubiquitous neural mechanisms mediating training-induced restoration in patients with long-standing V1 damage.

## Results

All eleven CB patients in the present study (see Table 1 for demographics) had long-standing ($35 \pm 68$ months, range: 5–237 months) unilateral, homonymous visual-field defects secondary to unilateral strokes affecting the occipital cortex or its immediate, post-chiasmatic afferents. Four had full hemianopia (CB5, CB7, CB9, CB11), five had partial hemianopia (CB1, CB3, CB4, CB8, CB10), and two sustained quadrantanopia (CB2, CB6). None of the patients were explicitly tested for the presence of blindsight abilities. Stroke-induced damage to the posterior cerebral artery and its territory resulted in lesions primarily located in the medial aspect of the occipital lobe and calcarine sulcus (average lesion volume: $16,673 \pm 18,592$ mm$^3$, range: 667–64,320 mm$^3$). The precise extent of the lesions into the cuneus and the lingual gyrus varied across patients, as did the degree of ventricular enlargement, suggesting differential involvement of the optic radiation and occipital white matter tracts. Anatomically, the occipital pole was intact in all cases, consistent with preserved foveal sensitivity ($35.7 \pm 0.85$ dB HVF sensitivity) and ability to fixate precisely (measured with eye-tracking).

**Training-induced recovery of visual discrimination and luminance detection sensitivity in CB fields.** All patients trained on visual discrimination tasks iteratively, over several months, at multiple blind-field locations (Fig. 1a, b and Table 1; mean ± SD training duration: $20.5 \pm 13$ months, range 3–46 months)[21–25]. CB patients were trained to discriminate visual stimuli presented at specific blind-field locations while maintaining fixation at the center of the screen. Note that the stimuli used for our testing and training had properties outside those known to elicit blindsight[4,24] (see "Methods"). Following training, chronic CB patients exhibited two types of partially-overlapping improvements: (1) improved visual discrimination performance at trained blind-field locations[21,23–25], and (2) recovered luminance detection sensitivity along the blind-field border, measured by

**Table 1 Patient demographics.**

| Patient | Sex | Age (years) | Damaged hemisphere | Lesion (mm3) | Time post-lesion (months) | Pre-post interval (months) | Pre-training deficit size (deg2) | | Post-training improvement (deg2) | | Training regime |
|---|---|---|---|---|---|---|---|---|---|---|---|
| | | | | | | | full HVF | inner 11.5° | full HVF | inner 11.5° | |
| CB1 | M | 67 | L | 6,422 | 10.4 | 5.5 | 693 (86%) | 133 (51%) | 122 | 40 | Double |
| CB2 | F | 58 | R | 16,303 | 11.8 | 10.8 | 294 (37%) | 112 (44%) | 318 | 85 | Static |
| CB3 | M | 63 | L | 667 | 38.4 | 34.4 | 473 (59%) | 100 (38%) | 66 | 51 | Double |
| CB4 | F | 29 | R | 2,126 | 5 | 3.1 | 489 (61%) | 129 (49%) | 49 | 44 | Static |
| CB5 | M | 72 | L | 10,574 | 11 | 9.7 | 764 (95%) | 229 (87%) | 57 | 7.4 | Double |
| CB6 | F | 68 | R | 64,320 | 26.2 | 16.9 | 194 (24%) | 42 (16%) | 134 | 50 | Motion |
| CB7 | F | 63 | L | 34,123 | 10.6 | 28.3 | 797 (99%) | 253 (96%) | 56 | 56 | Double |
| CB8 | M | 63 | L | 3,707 | 9.1 | 16.2 | 598 (74%) | 124 (47%) | 77 | 18 | Double |
| CB9 | F | 52 | R | 22,880 | 10.6 | 30.7 | 799 (99%) | 254 (96%) | 13 | 12 | Double |
| CB10 | M | 63 | L | 9,776 | 237 | 46.1 | 704 (87%) | 182 (69%) | 186 | 87 | Double |
| CB11 | M | 77 | L | 12,500 | 16.6 | 23.2 | 804 (99%) | 258 (98%) | 117 | 25 | Double |
| N = 11 | 5 F ∣ 6 M | 61 ± 13 | 7 L ∣ 4 R | 17 K ± 19 K | 35 ± 68 | 20.5 ± 13 | 601 ± 213 (75%) | 165 ± 74 (63%) | 109 ± 85 | 43 ± 27 | - |

All 11 patients suffered from unilateral, stroke-induced damage to the occipital cortex at least 5 months before training onset, causing a loss in luminance detection sensitivity assessable using Humphrey's Visual Field (HVF) mapping. Pre-training percent deficit size and post-training improvement area were computed for the full HVF and inner ±11.5 deg (i.e., area stimulated during fMRI). Following visual discrimination training (static orientation, motion direction, or both), all patients showed HVF improvements (i.e., area with ΔHVF ≥ + 6 dB).

automated HVF perimetry[21]–the gold standard, clinical method of assessing visual-field defects. Before training, CB patients were unable to discriminate the orientation (accuracy: $55.2 \pm 5.8\%$) or global motion direction (normalized thresholds: $86.2 \pm 12.2\%$) of visual targets presented within their blind field (Fig. 1). This was despite normal performance at corresponding locations within their intact field ($\text{intact}_{pre}$ vs $\text{blind}_{pre}$; orientation discrimination: $t(9) = 27.44$, $p < 0.001$, $d = 8.68$, Fig. 1c; motion thresholds: $t(8) = 20.67$, $p < 0.001$, $d = 6.89$, Fig. 1d). Following training, performance improved at trained locations ($\text{blind}_{pre}$ vs $\text{blind}_{post}$; orientation discrimination: $t(9) = 15.9$, $p < 0.001$, $d = 5.03$, Fig. 1c; motion thresholds: $t(8) = 16.00$, $p < 0.001$, $d = 5.33$, Fig. 1d), reaching levels not significantly different from those at intact-field locations ($\text{intact}_{post}$ vs $\text{blind}_{post}$; orientation discrimination: $t(9) = 1.64$, $p = 0.135$, $d = 0.52$, Fig. 1c; motion thresholds: $t(8) = 1.28$, $p = 0.235$, $d = 0.43$, Fig. 1d). Thus, chronic CB patients were able to relearn discrimination of stimuli not optimal for blindsight with intensive, visual discrimination training.

Critically for the present study, and as reported for a larger cohort of CB patients[21], visual discrimination training did not solely recover performance on trained tasks at trained, blind-field locations; it also improved conscious luminance detection (HVF) sensitivity across both trained and untrained blind-field regions[21] (Fig. 2 and Supplementary Fig. S1). CB fields are characterized by an abrupt fall-off in HVF sensitivity, from typically-intact (~30 dB) to perimetrically-blind (0 dB) levels of HVF sensitivity. For each patient, the blind-field border delimited impaired regions where the binocular average HVF sensitivity dropped below 15 dB in the initial (pre-training) HVF map (see *Methods*). Among the patients included here, the pre-training size of homonymous visual-field defects for the full HVF maps was $601 \pm 213$ deg$^2$ (~75% of the measured hemifield), ranging from 194 (24%) to 804 (99.8%) deg$^2$. In the central ±11.5 deg (i.e., the visual-field extent stimulated during fMRI), HVF deficits covered an area $165 \pm 74$ deg$^2$ (63% of the central ±11.5 deg hemifield), ranging from 42 (16%) to 258 (98%) deg$^2$. Following training, all CB patients showed recovery in HVF sensitivity within their blind field (Table 1), defined as a change ≥ +6 dB, which corresponds to double or more the test-retest variability of HVF measurements. HVF sensitivity improved by ≥ +6 dB over $109 \pm 85$ deg$^2$ across the full HVF fields (range: 13-318 deg$^2$; Supplementary Fig. S1) and over $43 \pm 27$ deg$^2$ (range: 7.4–86.5 deg$^2$) within the inner ±11.5 deg (Fig. 2). Little or no worsening (HVF decrements ≥ −6 dB) was observed (full HVFs: $-2 \pm 6.2$ deg$^2$; inner ±11.5 deg: $-0.9 \pm 2.6$ deg$^2$). Note that chronic CB patients do not exhibit significant *spontaneous* HVF recovery[1,2,21]. Significant HVF recovery occurs only in trained CB patients and directly correlates with the amount of training, whereas untrained, chronic CB patients show minimal changes in HVF, with small areas of improvement counterbalanced by areas of worsening[21]. Nonetheless, training-induced recovery of both HVF sensitivity and visual discrimination in chronic CB occurred preferentially along the blind-field border, consistent with the notion that this region may possess enhanced plastic potential[21].

To better understand the underlying mechanisms of training-induced recovery in luminance detection exhibited by chronic CB patients along their blind-field border, we directly compared changes in HVF sensitivity to retinotopic fMRI responses in the damaged V1 before and after training.

**Presence of spared representations of the blind field in perilesional V1 cortex before training**. Qualitatively, intact hemispheres of CB patients and visually-intact controls appeared similar, with V1 and most extrastriate cortical areas (e.g., V2, V3,

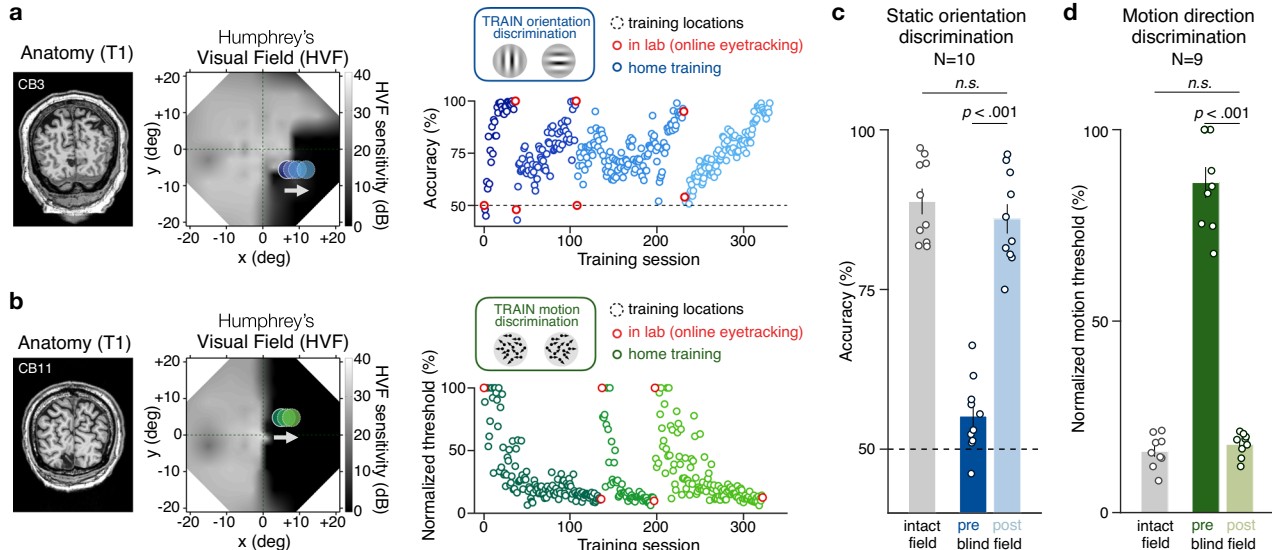

**Fig. 1 Visual discrimination training recovers visual functions in chronic cortically-blind (CB) fields. a, b** T1-weighted MRI and full (±21 deg) Humphrey Visual Field (HVF) for two chronic CB patients with stroke-induced damage to the primary visual cortex and associated loss of conscious luminance detection sensitivity. Dark regions (low HVF sensitivity) correspond to blind-field areas. Visual training can successfully restore (**a**) static orientation discrimination and (**b**) coarse (left/right) global motion direction discrimination in the blind field of chronic CB patients. Recovery is typically retinotopically-specific, requires weeks of daily home training (blue/green dots in scatter plots), and can be verified in lab under eye-tracking control (red dots in scatter plots). As detailed in previous work[21–25], all 11 CB patients included in the present study trained on (**c**) static orientation discrimination (N = 10) and/or (**d**) motion direction discrimination (N = 9) at different blind-field locations. All CB patients recovered performance levels on these tasks similar to those at equivalent locations in their intact visual hemifields. Bars show mean performance across subjects (±1 SEM), with individual data superimposed.

V4, hMT+) identifiable on retinotopic maps (Supplementary Fig. S2) as reported in previous studies[39,40]. The stroke-affected hemispheres of CB patients retained coarse retinotopic organization (Fig. 3). V1 was identified in all damaged hemispheres, using both phase reversals and anatomical markers (e.g., spared segments of the calcarine sulcus, comparison to the intact hemisphere; see *Methods*). Consistent with prior studies[33,38,41,42], the visual system of chronic CB patients retained visual representations of the blind field (Fig. 4a). Before training, a substantial portion of V1 voxels (22 ± 23%) of the damaged hemisphere represented blind-field regions with impaired HVF sensitivity (≤15 dB), with most of these voxels (16±17%) corresponding to locations with less than 6 dB HVF sensitivity (i.e., double the test-retest variability of HVF measurements). The presence of such preserved representations of the blind field within the damaged V1 has been speculated to be a potential neural substrate for training-induced recovery[11,21,29,38]. Here, we directly tested this hypothesis. To do so, we restricted our analyses to V1 voxels from the damaged hemisphere representing blind-field locations before training (pre-training HVF sensitivity: 0–15 dB).

The vast majority of V1 voxels representing regions of the blind field (N = 431 voxels) were in close proximity to the initial (pre-training) blind-field border (Fig. 4b). As expected, pre-training residual HVF sensitivity at these blind-field locations rapidly declined as we moved deeper into the blind field (Supplementary Fig. S3a). A decrease in V1 visually-evoked response was observed for spared V1 voxels representing blind-field locations further away from the blind-field border (Fig. 4c). Note that this drop in V1 responses was not due to the increased eccentricity of voxels further away from the blind-field border, as no drop is observed with eccentricity for V1 voxels in the intact hemispheres (Supplementary Fig. S3c) or in age-matched neurotypical participants (Supplementary Fig. S3d). Importantly, most training-induced HVF improvements occurred at blind-

field locations represented by V1 voxels near the blind-field border (Fig. 4d), which further supports the notion that this may indeed be a region of enhanced plastic potential, where repetitive stimulation with training–several months or even years post-stroke–can more readily recover luminance detection sensitivity.

**Pre-training, spared V1 representations of the blind field predict recovery in HVF sensitivity.** Critically, spared V1 representations measured prior to training using fMRI retinotopic mapping predicted the magnitude of post-training HVF recovery. We used a generalized linear mixed-effects model to predict the amount of post-training HVF recovery based on the properties of spared V1 voxels prior to training (see Methods). Our results reveal that spared V1 voxels with stronger pre-training, visually-evoked response coherence and representing blind-field regions in proximity to the blind-field border were associated with the greatest improvements in HVF recovery following training (Fig. 4e; generalized linear mixed-effects model with participants as a random effect: adj-$R^2$ = 0.55; intercept: $t = 21.38$, $p < 0.0001$, $\beta = 1.972$ [1.791 to 2.153]; pre-training V1 coherence: $t = 3.39$, $p = 0.0008$, $\beta = 0.673$ [0.283–1.063]; blind-field depth: $t = 2.52$, $p = 0.012$, $\beta = -0.072$ [−0.128 to −0.016]; interaction: $t = 3.18$, $p = 0.0016$, $\beta = -0.175$ [−0.282 to −0.066]). A similar pattern was observed when combining all voxels across patients, or when using pre-training response amplitude instead of coherence. Voxels closer to the blind-field border were associated with higher residual pre-training HVF sensitivity [though still abnormal], but this was only predictive of post-training HVF recovery when combining voxels across all participants (Supplementary Fig. S3e). Moreover, whether V1 voxels were closer to one of the discrimination-trained locations did not determine the amount of post-training HVF recovery observed (Supplementary Fig. S3f), consistent with our previous report[21]. It is worth noting that this analysis was limited by the

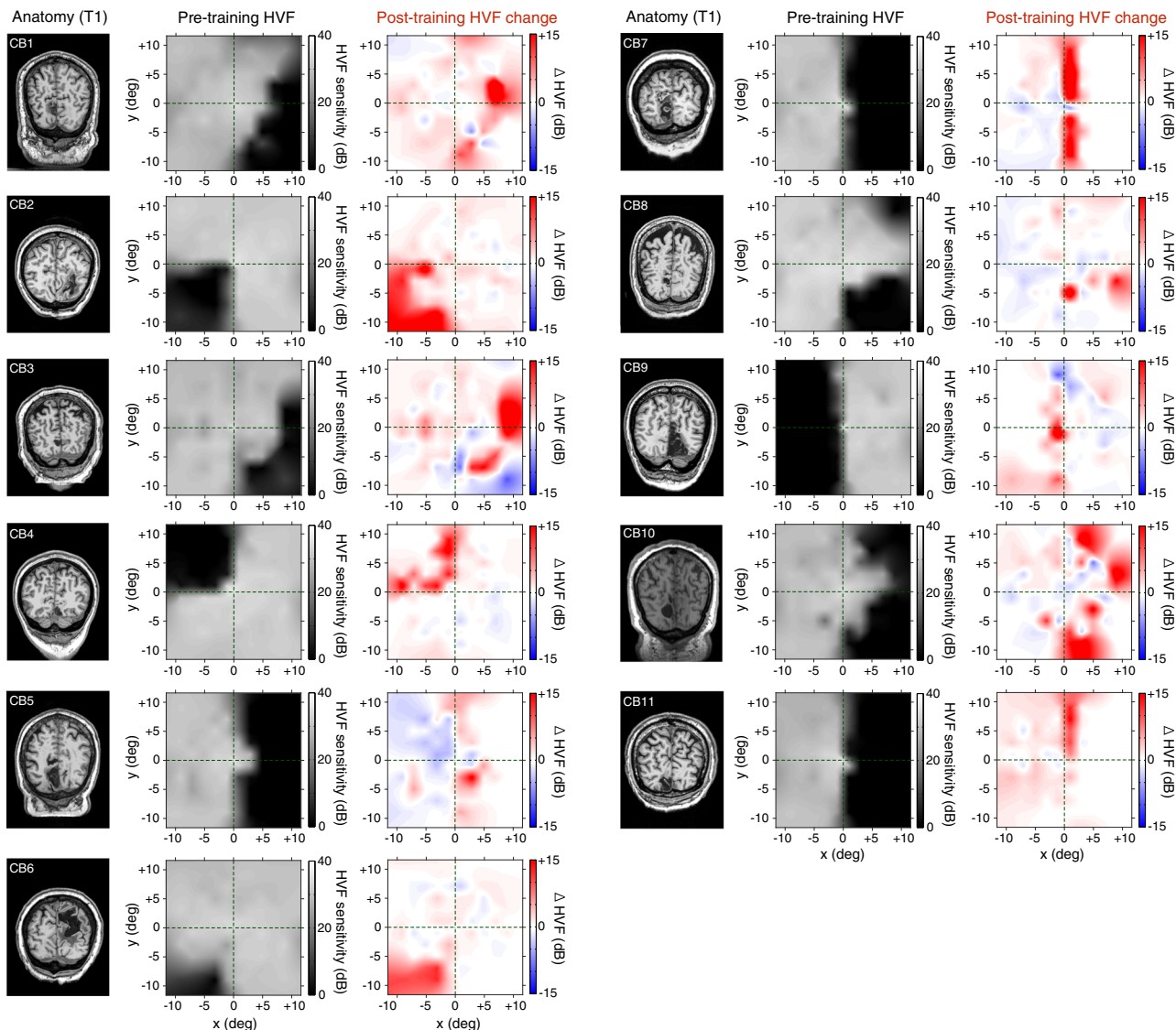

**Fig. 2 Visual discrimination training improves conscious luminance detection sensitivity in chronic CB patients.** T1-weighted MRI and corresponding baseline (pre-training) composite Humphrey Visual Fields (HVF; luminance detection sensitivity in dB) for all 11 chronic CB patients over the central 11.5 degrees of the visual field–i.e., the area stimulated during fMRI retinotopic mapping (see Supplementary Fig. S1 for individual maps of the full HVFs showing locations used for visual discrimination training). Prior to training, all CB patients showed homonymous loss of HVF sensitivity (dark regions) within parts of their visual field. As indicated by the post-training HVF change maps, all patients showed improved HVF sensitivity (red regions) following training, which was greatest within the confines of their initial (pre-training) blind-field border. A substantial amount of this recovery occurred within the inner 11.5 degrees of the visual field (see[21] for full description of training-induced HVF recovery in a larger cohort of chronic CB patients). Areas of sensitivity loss (blue) were also noted in some patients, but only 2 (CB3 and CB9) exhibited significant loss ($\geq$-6dB) along with larger areas of HVF improvements.

fact that training locations for nearly half the patients were located outside the scanning area (see Supplementary Fig. S1).

Additional regression analyses showed that pre-training V1 activity alone was predictive of the magnitude of HVF recovery observed following training, both in terms of pre-training V1 response coherence (Fig. 4f; adj-$R^2 = 0.25$; intercept: $t = 12.78$, $p < 0.0001$, $\beta = 1.66$ [1.40–1.91]; slope: $t = 3.35$, $p = 0.0009$, $\beta = 0.863$ [0.357–1.370]) and amplitude (Fig. 4g; adj-$R^2 = 0.27$; intercept: $t = 26.23$, $p < 0.0001$, $\beta = 1.81$ [1.68–1.95]; slope: $t = 2.54$, $p = 0.0114$, $\beta = 0.35$ [0.080–0.625]). The depth in the blind field of V1 voxels representing blind-field locations was also predictive of the magnitude of post-training HVF recovery, with greater HVF recovery being associated with V1 voxels representing blind-field locations closer to the blind-field border (Fig. 4h; adj-$R^2 = 0.53$; intercept: $t = 23.88$, $p < 0.0001$, $\beta = 2.29$ [2.10–2.48]; slope: $t = 8.83$, $p < 0.0001$, $\beta = -0.159$ [−0.194 to

−0.123]). Taken together, these results support the notion of enhanced plastic potential in perilesional V1 cortex of CB patients, even years after stroke-induced cortical damage. Critically, the *strength* of spared V1 activity representing perimetrically-blind locations along the blind-field border prior to training predicts greater recovery of conscious luminance detection sensitivity following training.

**Subtle post-training differences in spared V1 responses.** Of the 11 CB patients scanned pre-training, only 8 were able to complete the post-training fMRI session (CB1-8; see "Methods"). Overall, pre- and post-training retinotopic maps were qualitatively similar, suggesting no global cortical reorganization following training (Fig. 5a; see Supplementary Fig. S4 for individual maps). To evaluate changes in retinotopic V1 representations associated with training-induced HVF recovery, we analyzed visually-evoked

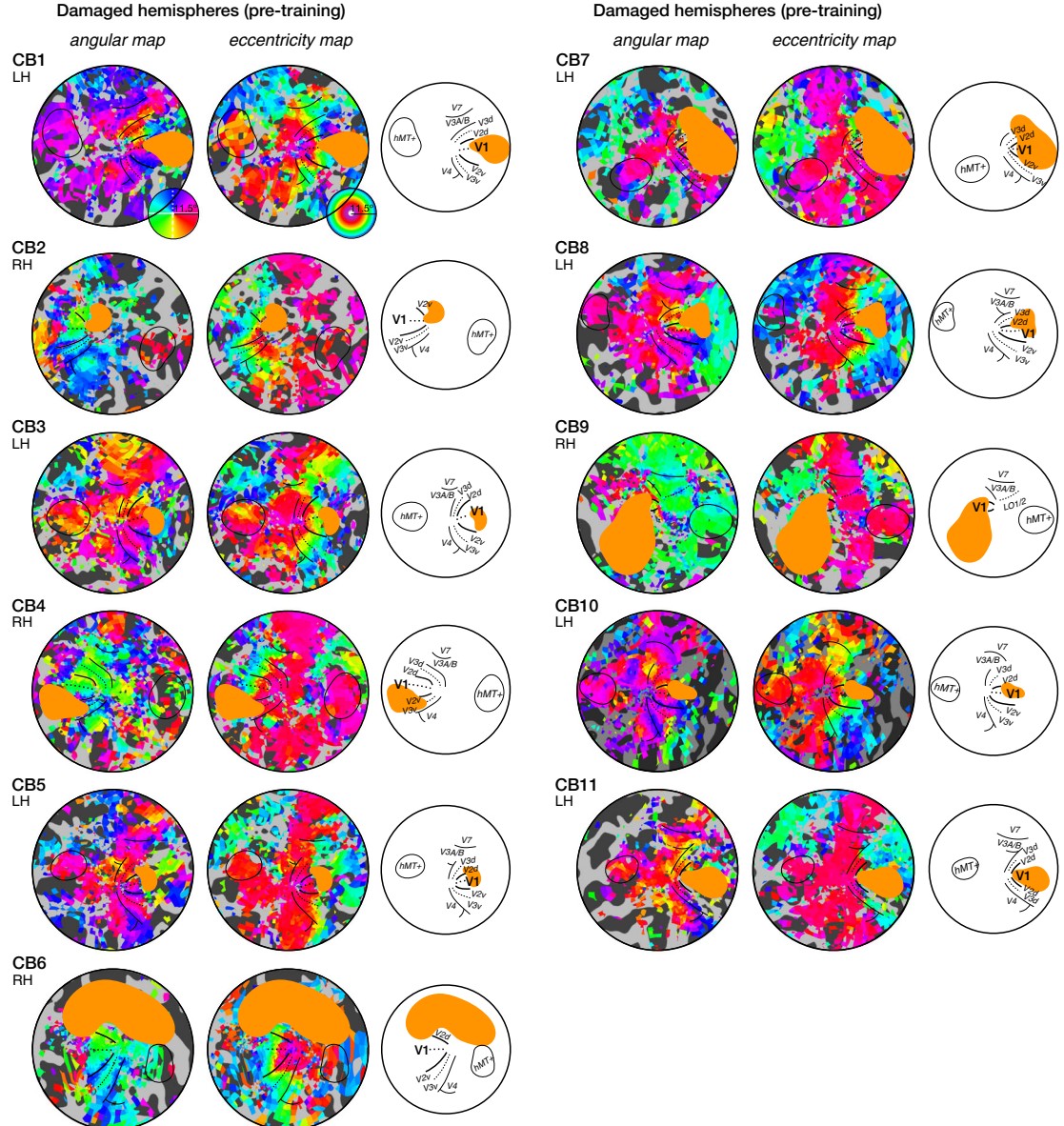

**Fig. 3 Pre-training retinotopic maps of the damaged hemispheres of all 11 chronic CB patients.** All patients retained retinotopic organization, both in terms of radial and eccentric representations. Stroke-induced lesions (orange masks) overlapped with V1 in most cases, as well as with other extrastriate areas.

responses of V1 voxels representing regions of the initial (pre-training) blind field separately for pre- and post-training sessions (594 voxels in total; 297 pre-training, 297 post-training). We used a generalized linear mixed-effects model, with participants as a random effect, to assess post-training changes in visually-evoked V1 responses (coherence or amplitude) as a function of the magnitude of post-training HVF recovery. As observed pre-training, stronger V1 response coherence was associated with greater HVF recovery (adj-$R^2 = 0.60$; intercept: $t = 5.92$, $p < 0.0001$, $\beta = -1.08$ [$-1.43$ to $-0.72$]; slope with HVF change: $t = 3.19$ $p = 0.0015$, $\beta = 0.043$ [0.016–0.069]), but training did not affect V1 response coherence nor interacted with the level of HVF recovery observed at these blind-field locations ($p$-values > 0.1) (Fig. 5b). With respect to response amplitude (Fig. 5c), we observed a difference post-training: V1 voxels representing blind-field regions that showed no or weak HVF recovery post-training were associated with larger response amplitude following training, whereas blind-field regions with greater HVF recovery did

not show a consistent change (adj-$R^2 = 0.25$; intercept: $t = 3.96$ $p < 0.0001$, $\beta = -0.78$ [$-1.17$ to $-0.39$]; HVF recovery: $t = 3.00$ $p = 0.0028$, $\beta = 0.05$ [0.017–0.080]; training: $t = 2.22$, $p = 0.0268$, $\beta = 0.364$ [0.042–0.685]; interaction: $t = 2.89$, $p = 0.0040$, $\beta = -0.065$ [$-0.110$ to $-0.021$]). In addition, training did not result in a consistent change in the preferred position of V1 voxels relative to the blind-field border; i.e., the depth in the blind field (Fig. 5d; adj-$R^2 = 0.47$; intercept: $t = 2.42$ $p = 0.016$, $\beta = 0.82$ [0.15–1.49]; HVF recovery: $t = 2.10$ $p = 0.0368$, $\beta = -0.10$ [$-0.193$ to 0.006]; training: $t = 0.93$, $p = 0.351$, $\beta = -0.08$ [$-0.25$ to 0.09]; interaction: $t = 1.61$, $p = 0.109$, $\beta = 0.028$ [$-0.006$ to 0.063]). No difference was observed for ipsilesional V1 voxels covering perimetrically intact regions, or for V1 voxels of the intact hemisphere. A similar pattern was observed when combining voxels across all 8 patients. In summary, training-induced HVF recovery occurred at blind-field locations with strong, pre-training V1 activity, which did not change further following training. If anything, training enhanced visually-evoked V1

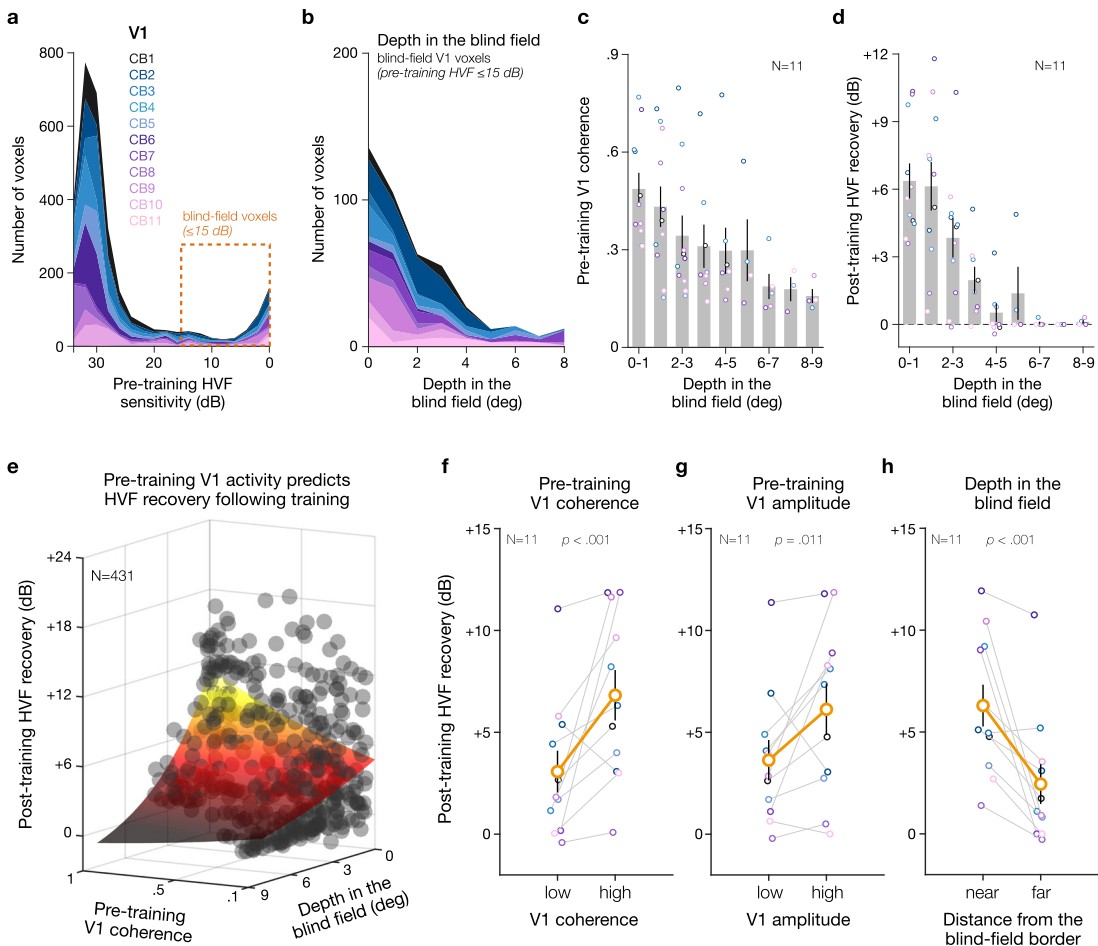

**Fig. 4 Spared pre-training V1 activity predicts post-training Humphrey's Visual Field (HVF) recovery in the blind field of chronic CB patients. a** Distribution plot of V1 voxels in damaged hemispheres as a function of the pre-training HVF sensitivity (using 2 ± 1 dB steps). **b** Distribution plot of V1 voxels in damaged hemispheres representing regions of the blind field (≤15 dB pre-training HVF sensitivity) prior to training as a function of the depth in the blind field (i.e., distance from the blind-field border) using 1 ± 0.5 deg steps. Each color corresponds to a single CB patient. Note the preferential location of these voxels near the blind-field border (i.e., 0 deg). **c**, **d** Consistent with enhanced plastic potential along the blind-field border of CB patients, spared blind-field locations closer to the blind-field border showed stronger visually-evoked V1 response coherence prior to training, as well as greater HVF change (in dB of sensitivity) from pre- to post-training. Bars represent average estimates across CB patients (±1SEM), with individual dots corresponding to individual CB patients. **e** The strength of pre-training visually-evoked responses and the blind-field depth of V1 voxels representing blind-field locations were predictive of the magnitude of post-training HVF recovery, with a significant interaction between pre-training V1 coherence and depth in the blind field. Each data point corresponds to a V1 voxel, which were fit (colored surface) using a generalized linear mixed-effects model with participants as a random effect (adj-r2 = .55). Post-training HVF recovery as a function of pre-training, visually-evoked V1 responses within the blind field (**f**: coherence; **g**: amplitude) computed for each CB patient (N = 11). Data were split using the group median for illustration purposes only (coherence: 0.37; amplitude: 0.54); p-values were computed from the generalized linear mixed-effects analyses on unbinned data. Error bars correspond to ±1SEM, with gray lines representing individual subjects. (**h**) Same as (**f**, **g**) but as function of the blind-field depth (median split: 1.7 deg).

response amplitude for blind-field locations with weaker, pre-training activity, without resulting in an associated, significant increase in HVF sensitivity.

**Increased pRF size in perilesional V1 cortex following training.** To further characterize changes in V1 associated with training-induced HVF recovery, we used the population receptive field (pRF) method[43,44] to estimate the position and size of the visual-field area that best explained each voxel's visually-evoked response. For each observer, we assessed how pRF properties changed with training, as a function of whether V1 pRFs covered regions with impaired pre-training HVF sensitivity, or covered only regions with preserved HVF sensitivity (Fig. 6; see "Methods"), excluding pRFs with less than 10% variance explained (r²). Consistent with the presence of spared, phase-encoded V1 responses within the blind-field (Figs. 3, 4), a substantial number

of V1 pRFs (18 ± 15%) covered parts of the blind field prior to training, providing further evidence that perilesional V1 cortex in chronic CB patients retains spared representations of the blind field[33,38,41,42]. Two-way repeated-measures ANOVAs on pRF estimates (Fig. 7 and Supplementary Fig. S5) indicated no effect of training (pre- vs. post-training), or interaction between training and HVF coverage (blind-field vs. intact HVF) on the variance explained (r²), preferred eccentricity, position depth in the blind-field, or the number of V1 pRFs (all p > 0.1). As expected, there were fewer V1 pRFs covering blind-field regions than pRFs covering visual-field locations with preserved HVF sensitivity (Supplementary Fig. S5c; F(1,7) = 32.95, p < 0.001, $\eta_p^2$ = 0.83). Overall, blind-field pRFs had a similar variance explained (Supplementary Fig. S5a; F(1,7) = 2.22, p = 0.180, $\eta_p^2$ = 0.24), were by definition located near the blind-field border (Fig. 7b; F(1,7) = 61.32, p < 0.001, $\eta_p^2$ = 0.9), and were both more

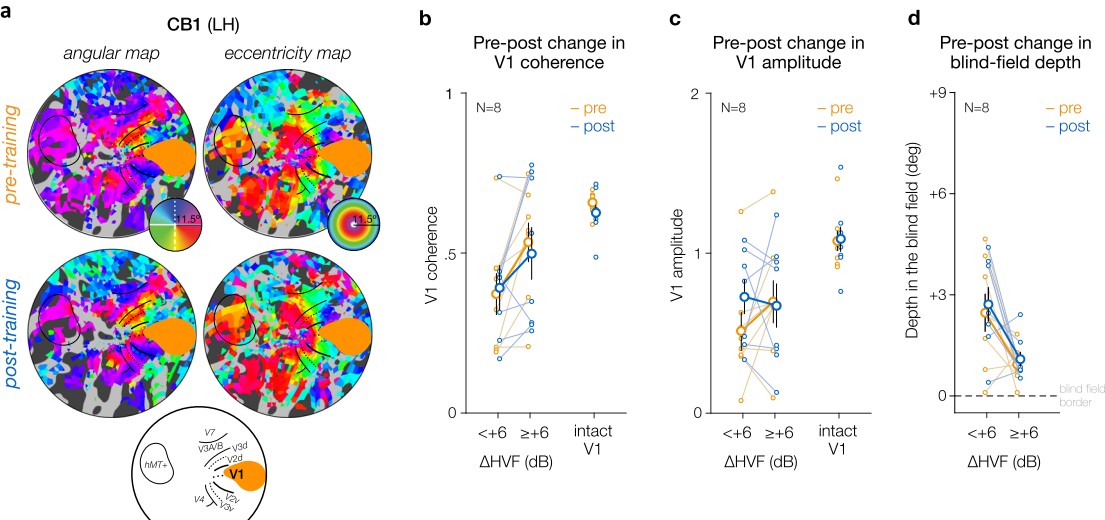

**Fig. 5 Retinotopic organization and V1 activity in CB patients ($N = 8$) following training. a** Sample pre-training and post-training retinotopic maps of the damaged hemispheres for one of our CB patients (see Supplementary Fig. S4 for all individual post-training maps). Qualitatively, no global change in retinotopic organization was observed following training. V1 response (**b**: coherence; **c**: amplitude) for voxels representing blind-field locations prior to training, are then plotted as a function of training and post-training HVF recovery. Although no consistent change in V1 coherence was observed following training, V1 voxels representing blind-field locations with weak HVF recovery showed increased response amplitude. Data were split based on the level of HVF recovery for illustration purposes only (no recovery: ΔHVF < + 6 dB; significant recovery: ΔHVF ≥ + 6 dB), while generalized linear mixed-effects models on unbinned data, with participants as a random effect, were used for statistical analyses. Solid symbols represent group-averaged values with ±1SEM error bars, with each individual CB patient represented by thinner lines. **d** Same as (**b**, **c**) but for the depth in the blind field of the preferred position of V1 voxels before and following training as a function of the amount of HVF recovery observed at that blind-field location.

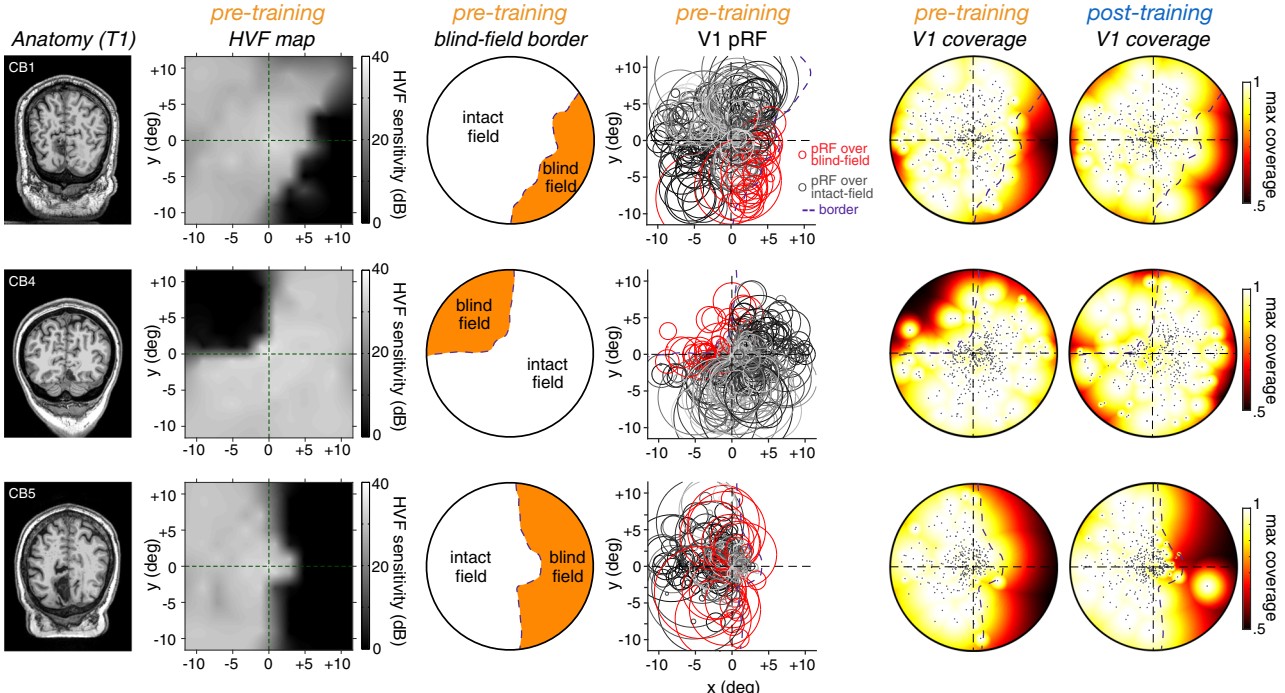

**Fig. 6 Population receptive field (pRF) and visual field coverage before and following training.** Anatomy (T1), pre-training HVF map, pre-training V1 pRFs, and resulting visual-field coverage maps (maximum coverage) for pre-training and post-training V1 pRFs shown for 3 CB patients. The pRF method was used to estimate the position and size of the visual-field area that best explained each voxel's visually-evoked response. The initial (pre-training) HVF map was used to define the blind-field border and determine whether each single pRF covered blind-field regions or solely intact-field regions with preserved HVF sensitivity. Single black dots on the coverage maps indicates the preferred center of each V1 pRFs.

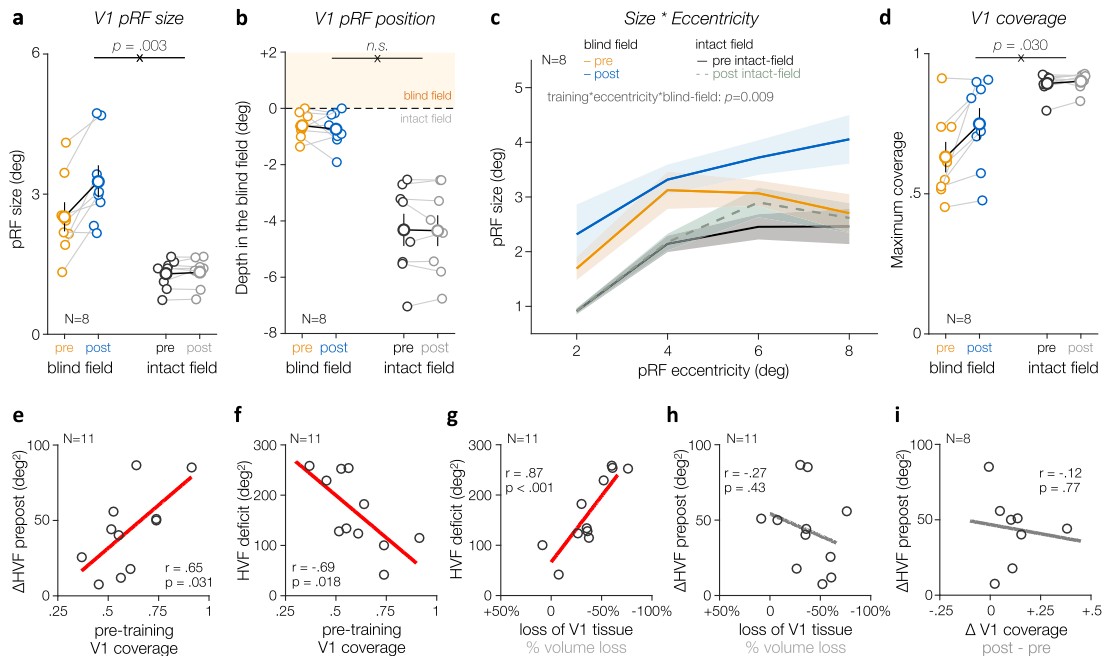

**Fig. 7 Enhanced population receptive field size and coverage of the blind field in V1 following training. a** Training in CB patients ($N = 8$) resulted in a significant increase in pRF size for V1 pRFs covering blind-field regions, compared to V1 pRFs covering solely intact-field regions. Solid symbols show group-averaged estimates (±1SEM), and smaller dots individual data. Significant pRF coverage*training interaction (repeated-measures ANOVA) is indicated above the graph. **b** Training was not associated with a change in V1 pRF preferred position relative to the blind-field border, expressed as the depth in the blind field (in deg). Same convention as in (**a**). Moreover, training was not associated with changes in variance explained, pRF eccentricity or number of V1 pRFs (see Supplementary Figs. S5 and S6). **c** Prior to training, pRF size plotted as a function of eccentricity indicated larger pRF size for V1 pRFs covering parts of the blind field, compared to V1 pRFs covering the intact field. This difference decreased with eccentricity. Following training, pRF covering the blind field increased in size, with this increase being more pronounced with eccentricity. Shaded areas correspond to ±1SEM. Data were binned as a function of eccentricity (2–8 deg with 2 ± 1 deg steps) for illustration purpose only; effects were assessed using a linear mixed-effects model. **d** Increase in the size of perilesional V1 pRFs resulted in enhanced V1 coverage of the blind field following training. Same convention than (**a**, **b**). **e** Pre-training V1 coverage of the blind field in CB patients ($N = 11$) was predictive of the amount of HVF recovery observed within the inner 11.5 deg following training. The initial HVF deficit area (for the inner 11.5 deg) was anti-correlated with (**f**) the amount of pre-training coverage of the blind field and (**g**) with the loss of V1 tissue (i.e., negative values indicate a stronger reduction in V1 volume between the intact and damaged hemispheres). **h** However, the loss of V1 tissue (and the size of the pre-training HVF deficit) did not correlate with the amount of HVF recovery observed following training. **i** Larger increase in V1 coverage of the blind field (or in the size of V1 pRFs) was not associated with stronger HVF recovery.

eccentric (Supplementary Fig. S5b; $F(1,7) = 26.66$, $p < 0.001$, $\eta_p^2 = 0.79$) and larger in size (Fig. 7a; $F(1,7) = 32.53$, $p < 0.001$, $\eta_p^2 = 0.82$) than the latter.

Critically, we found a significant effect of training on pRF size ($F(1,7) = 20.33$, $p = 0.003$, $\eta_p^2 = 0.74$), with a significant interaction between training and HVF sensitivity ($F(1,7) = 19.13$, $p = 0.003$, $\eta_p^2 = 0.73$)(Fig. 7a). These effects reflected a significant enlargement of V1 pRFs covering the blind field by ~+34% following training ($t(7) = 4.54$, $p = 0.003$, $d = 1.60$), whereas there was no significant change in size for pRFs covering regions with preserved, baseline HVF sensitivity ($t(7) = 0.91$, $p = 0.393$, $d = 0.32$). Note that the increase in pRF size over blind-field regions post-training was not associated with a consistent change in pRF position relative to the blind-field border (Fig. 7b and Supplementary Figs. S6 and S7). To further characterize this post-training enlargement of V1 pRFs covering the blind field, we used a linear mixed-effect model to predict differences in pRF size as a function of pRF eccentricity, HVF coverage, and training, using participants as a random effect and pRFs covering solely intact visual field regions in the pre-training condition as reference (Fig. 7c). As expected, pRF size covering intact-field locations prior to training increased with eccentricity (adj-$R^2 = 0.42$; intercept: $t = 6.44$, $p < 0.0001$, $\beta = 0.58$ [0.40–0.76]; eccentricity: $t = 5.07$, $p < 0.0001$, $\beta = 0.281$ [0.172–0.390]). There was neither a main effect of training nor an interaction between training and

eccentricity for pRFs covering intact-field locations ($p$ values > 0.1). Consistent with previous results in untrained, chronic CB patients[38], we found V1 pRFs covering blind-field regions to be significantly larger than pRFs covering solely intact-field regions at corresponding eccentricities prior to training (effect of HVF coverage: $t = 3.72$, $p = 0.0002$, $\beta = 2.39$ [1.13–3.65]), with this pre-training difference being attenuated with pRF eccentricity (interaction: $t = 3.33$, $p = 0.0009$, $\beta = -0.283$ [$-0.450$ to $-0.117$]). Following training, the size of V1 pRFs covering blind-field regions increased significantly, and this enlargement was more pronounced with eccentricity (three-way interaction: $t = 2.60$, $p = 0.0092$, $\beta = 0.186$ [0.046–0.327]). Thus, V1 pRFs covering the blind field were atypically large before training[38] and became even larger following training.

**Increased V1 coverage of the blind field following training.** Finally, we derived visual-field coverage maps over the central 11.5 deg by superimposing all V1 pRFs and computing the coverage index for regions with either preserved or impaired HVF sensitivity (Fig. 6 and "Methods"). Consistent with the post-training enlargement of pRF size over blind-field areas (Fig. 7a, c), training increased V1 coverage of the blind-field (Fig. 7d; training*HVF coverage interaction: $F(1,7) = 7.41$, $p = 0.030$, $\eta_p^2 = 0.51$). Regions with impaired HVF sensitivity were associated

with reduced visual-field coverage by V1 pRFs ($F_{(1,7)}=15.85$, $p = 0.005$, $\eta_p^2 = 0.69$). Visual-field locations with preserved HVF sensitivity were well covered by V1 pRFs, with no difference between testing sessions (pre: $89.4 \pm 4.1\%$; post: $90.2 \pm 3.2\%$; $t_7 = 1.11$, $p = 0.304$ d = 0.39), and were similar to visually-intact age-matched controls ($88.3 \pm 2.7\%$; $F_{(1,15)} = 0.14$, $p = 0.711$). Prior to training, CB patients showed overlap between pRF V1 maps and perimetrically-blind regions (Fig. 7d; pre-training coverage index: $63.1 \pm 15.4\%$, range: 36.8–91.3%). Following training, V1 coverage over blind-field areas improved (Wilcoxon signed-rank test: $p = 0.016$, $r = 0.944$), while no change was found for pRFs representing intact regions of the visual field ($p = 0.304$).

Notably, we found that pRF mapping can serve as a simple predictor of the recovery potential of chronic CB patients prior to training. Indeed, the amount of spared, pre-training V1 coverage of the blind field was significantly correlated with the area of HVF improvement observed following training within the central $\pm 11.5$ deg of the visual field (Fig. 7e; $r = +0.647$, $p = 0.031$). A similar pattern was observed when using the area of HVF improvement over the full HVF map ($r = +0.650$, $p = 0.030$), or when computing coverage maps using pRFs weighted by their own variance explained (HVF recovery over full field: $r = 0.59$, $p = 0.058$; for inner 11.5 deg: $r = 0.51$, $p = 0.11$). Note that whereas the pre-training deficit area estimated from HVF maps for the inner 11.5 deg area was anti-correlated with the amount of pre-training V1 coverage of the blind field (Fig. 7f; $r = -0.694$, $p = 0.018$), the area of pre-training HVF deficit was not a good predictor of the amount of HVF improvements observed post-training ($r = -0.365$, $p = 0.269$). Similarly, the loss of V1 tissue, estimated as the percent change in V1 volume between the damaged and intact hemispheres, was significantly correlated with the size of the pre-training HVF deficit (Fig. 7g; $r = +0.87$, $p = 0.0005$), and associated with reduced pre-training coverage of the blind-field ($r = -0.59$, $p = 0.054$). However, the amount of training-induced HVF recovery was neither correlated with the loss of V1 tissue (Fig. 7h; $r = -0.27$, $p = 0.427$) nor with the size of the initial HVF deficit ($r = -0.37$, $p = 0.269$). Thus, HVF deficits in chronic CB patients are related to the loss of V1 tissue, but the latter does not by itself determine the potential for training-induced HVF recovery.

These results suggest that spared V1 representations within perimetrically-blind regions of the visual field mediate recovery of conscious, luminance-detection sensitivity following training. Moreover, they detail what level of functionality these V1 circuits need to support training-induced improvements. As such, while the amount of training-induced HVF recovery did not correlate with the increase in blind-field coverage post-training (Fig. 7i; $r = -0.123$, $p = 0.772$), we speculate that the improvement in coverage of the blind field by V1 pRFs nonetheless plays a role in visual recovery, representing an increase in V1 responsiveness needed to support further recovery deeper within the blind field of CB patients once additional training would be administered.

**Limited post-training changes in extrastriate areas.** Although pre-training coverage of the blind field was also observed in extrastriate areas (V2-V4), these areas did not show a significant increase in pRF size (Fig. 8a–c) or in blind-field coverage (Fig. 8d–f) post-training. Similar trends were observed, but the differences in pRF size and in coverage of the blind field were smaller and may have been driven by patients who also showed V2/V3 damage (e.g., CB4). Moreover, similar levels of pre-training coverage of the blind field were observed in V1-V3. The overall decrease in visual-field coverage in V4 likely reflected artifacts

from the transverse sinus that mask fMRI responses on the ventral surface of the human visual cortex, resulting in a partial coverage of the contralateral hemifield[45]. Importantly, neither the *strength* of pre-training, visually-evoked responses (not shown) nor the amount of pre-training coverage of the blind field (Fig. 8g–j) were predictive of the amount of post-training HVF recovery for extrastriate areas. In addition, while pRFs covering the blind field prior to training were atypically large in V1 (Fig. 7c), this difference was substantially reduced in extrastriate areas (Supplementary Fig. S8), as previously observed in untrained chronic CB[38]. We note here that eye movements during fMRI acquisition are unlikely to explain our results (i.e., changes in V1 without clear changes in other areas), as it would have negatively impacted the quality of retinotopic maps and increased the size of all pRFs similarly across the visual field and across all visual areas[46]. Although these findings could suggest a tighter link between V1 retinotopic activity and the recovery of luminance detection sensitivity in chronic CB patients, we cannot rule out the presence of changes in extrastriate areas that might not have been captured by our retinotopic approach, but could be observed using other fMRI approaches (e.g., by measuring training-induced changes in BOLD activity for stimuli presented at specific blind-field locations and on which patients perform a task).

## Discussion

In the present study, we used fMRI and retinotopic mapping to examine the neural substrates of training-induced recovery of luminance detection sensitivity in CB patients with large, chronic, homonymous visual-field defects due to stroke-induced V1 damage. Six main findings emerged: (1) prior to training, chronic CB patients exhibited substantial, visually-evoked fMRI responses in perilesional V1 cortex corresponding to blind regions of their ipsilesional hemifields; (2) the strength of spared, pre-training V1 activity representing blind-field regions in close proximity to the blind-field border was predictive of the amount of luminance detection sensitivity recovered post-training; (3) pre-training coverage of the blind field in V1 (but not in V2–V4) was also predictive of the amount of training-induced HVF recovery; (4) following training, blind-field locations exhibiting greatest HVF recovery were not associated with changes in visually-evoked response; instead, training seemed to enhance V1 response amplitude at blind-field locations with weak pre-training V1 activity (and limited HVF improvements); (5) training-induced recovery was associated with increased V1 coverage of the blind field, mediated by an enlargement in the size of V1 pRFs covering blind-field regions; (6) although pre-training coverage of the blind field was also observed in extrastriate areas (V2-V4), no clear changes in pRF properties or visual coverage were observed in these areas following training. Taken together, our findings provide key insights regarding the neural mechanisms by which daily visual discrimination training administered over several months restores HVF sensitivity in the blind field of chronic CB patients. Above all, our results provide empirical evidence that spared V1 circuits representing blind-field regions serve as critical substrates of training-induced recovery of conscious luminance detection sensitivity in chronic CB.

**Presence of spared blind-field representations in perilesional V1 cortex of chronic CB patients.** Most prior neuroimaging studies on the properties of cortical reorganization and residual visual processing in CB involved case reports[33,35,47–52]. Only more recent investigations have examined larger patient cohorts[19,34,38,53–55]. Moreover, previous studies often faced interpretational issues that arose from significant inter-individual variability in the type, size, and age of cortical damage, among

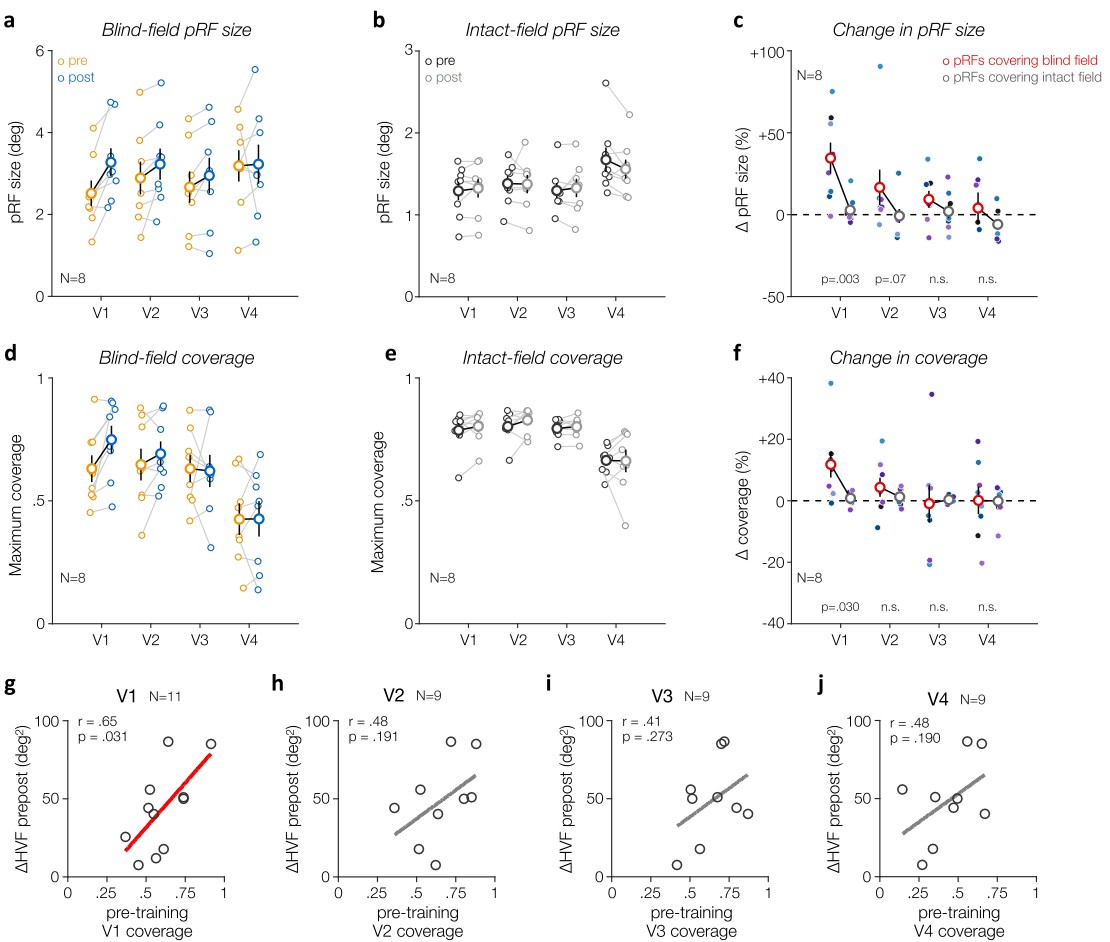

**Fig. 8 Lesion-induced and training-induced differences in pRF size and coverage in V1 but not in extrastriate areas (V2, V3, V4). a** V1 pRFs covering blind-field regions showed an increase in pRF size following training, which was not observed in extrastriate areas (V2-V4), or for (**b**) pRFs covering solely intact-field regions. **c** Summary of the changes in pRF size as a function of visual areas and whether pRFs covered blind-field or intact-field regions, computed as a percent change. **d, e** Training-induced enlargement of V1 pRFs covering regions of the blind field significantly increases coverage of the blind field in V1, without clear change in extrastriate areas (V2-V4) or for the coverage of the intact field. In panels a-f, small symbols correspond to individual CB patients ($N = 8$), with larger symbols corresponding to group-average values ($\pm 1$SEM). Statistical values for the training*HVF coverage interaction are provided at the bottom of (**c**) and (**f**). **g-i** Pre-training pRF coverage of the blind field was predictive of the amount of HVF change in V1, but not in extrastriate areas. Note that extrastriate areas could not be defined in two patients (CB9 and CB11), resulting in a different number of patients for extrastriate areas ($N = 9$) than for V1 ($N = 11$) in panels g-j (excluding these two patients from V1 analyses as well did not affect our findings).

others. Here, we restricted our recruitment to patients with visual impairments resulting from unilateral, chronic, stroke-induced V1 damage sustained during adulthood. The intent was to minimize inter-subject variability from diversity in underlying pathology (e.g., stroke *versus* resection or trauma), and from changes due to increased plasticity from damage early in development. Like prior work[33,36,38,47,51], we found that V1 damage did not induce large-scale reorganization of phase maps. Despite their longstanding stroke, most CB patients in the current study showed differentiable V1/V2 borders and phase reversals between most extrastriate visual areas. Importantly, we found preserved, perilesional V1 cortex representing regions of the perimetrically-blind visual field, consistent with prior studies on smaller groups of CB patients[33,38,41,42,52]. Note that residual visual representations within regions of the blind field in untrained chronic CB patients differ from controls tested with artificial scotomas, who only show limited visually-evoked activity and pRF coverage of masked visual-field locations, without pRF enlargement along the border of the blind field[38,56,57].

**Spared V1 activity prior to training predicts training-induced visual recovery.** The existence of residual, visually-evoked activity representing blind-field regions in the damaged V1 of CB patients has important implications, both for understanding preserved vision and designing better visual restitution strategies[7–9,11,18,27,29,31,38,42,58]. V1 plays a central role in the visual cortical hierarchy, but its exact contribution to visual awareness is still a matter of debate. Fluctuations in V1 activity are thought to reflect fluctuations in conscious perception[3,59–61]. Furthermore, V1 is the primary source of excitatory extrastriate input; as such, its damage alters cortical activity throughout the rest of the visual hierarchy—a factor thought to contribute to the loss of awareness[3,62]. Yet, there is also compelling evidence of decoupling between awareness and V1 activity[3,62,63]. For example, patients with complete unilateral or even bilateral V1 absence can still show forms of perceptual awareness triggered by visual stimuli presented in their blind field[64,65]. Thus, while not being a gatekeeper for visual awareness, V1 provides important conditions for awareness to arise. An important point relevant to the existence of spared V1 representations of HVF-defined blind regions of the visual

field is that in spite of this spared activity, chronic CB patients do not spontaneously recover either motion perception or luminance detection sensitivity (i.e. HVF perimetry) without training[1,2,21]. Thus, the mere presence of spared V1 activity representing portions of the blind-field is not sufficient for recovery to occur; deliberate training is required to restore conscious luminance detection within chronic CB fields. Although it is unclear why regions with preserved V1 activity do not mediate conscious perception without training in chronic CB patients, here we reveal the potential relevance of such preserved activity to visual restoration efforts. Specifically, the strength of pre-training fMRI activity representing perimetrically-blind regions in the affected V1, as well as the amount of pre-training V1 coverage of the blind field, predicted the magnitude of HVF recovery following training. This finding indicates that fMRI can provide vital insights into a patient's potential for training-induced recovery. In the future, more efficient rehabilitation strategies could utilize such information to optimize training by pre-selecting visual-field regions most amenable to recovery.

**Impact of training on V1 activity**. Our results show that blind-field locations with the greatest HVF recovery did not exhibit further increases in visually-evoked BOLD responses post-training. This finding may suggest that training-induced recovery in HVF sensitivity could be mediated by improvements in the read-out efficiency of sensory signals at blind-field locations with strong, initial (pre-training) V1 activity. Such a learning mechanism would be consistent with observations in intact visual systems, where improved behavioral performance with training often involves changes in sensory read-out efficiency at later decoding stages, rather than in early sensory representations[66–68]. Of special relevance given our patients' training regime, training-induced improvements in motion discrimination in intact (non-human primate) visual systems are not associated with enhanced firing rates in motion-sensitive area MT. Instead, they are associated with changes in firing rates in the lateral intraparietal cortex (LIP), which is implicated in accumulating sensory evidence[68]—something that appears to be impaired in CB fields[53]. Indeed, a drift diffusion model of decision making applied to fMRI responses revealed slower rates of information accumulation (i.e., reduced drift rate) when untrained CB patients performed a global direction discrimination task at blind-field locations, relative to their intact fields and to intact controls[53]. Unfortunately, the retinotopic approach used presently was not optimal nor designed to measure task- and modality-specific changes in visual processing. The role and possible changes in read-out efficiency in other brain areas would be better characterized by measuring brain activity at different stages of training while CB patients are asked to detect and/or discriminate visual information at specific locations within their blind field.

Although our results suggest plasticity at neural decoding rather than encoding stages, we did observe post-training changes in V1 representations, specifically for regions deeper in the blind-field than those exhibiting HVF recovery. These blind-field locations had very weak pre-training V1 activity, but showed enhanced visual field coverage and to a certain extent, increased visually-evoked response amplitude in V1 following training. The origin and functional significance of changes observed deeper within the blind field–where patients remained perimetrically blind following this initial phase of training–remain to be elucidated. We hypothesize that a certain level of baseline V1 responsiveness is necessary for training to effectively induce recovery of conscious luminance detection. In this context, locations that remained perimetrically blind but showed enhanced post-training V1 activity might serve as a new *baseline* upon which further training could act to induce additional HVF

recovery. Such a "bootstrapping" mechanism would be consistent with the observation that sequential, iterative training can recover vision progressively deeper within chronic CB fields[21,24,25]. Thus, while such changes were too small to mediate recovery of conscious luminance detection sensitivity (consistent with our earlier finding), they may reflect changes in unconscious visual abilities (i.e., blindsight) post-training. As we did not test for blindsight in our patients, we can only speculate about its role in this context. Both our fMRI and psychophysical results were consistent across patients hereby presented, suggesting that initial differences in blindsight abilities, if present, had little impact on the amount of HVF recovery and changes in perilesional V1 activity attained. It is noteworthy that repeated stimulation in chronic CB patients has been shown to affect the type of blindsight observed[16,27,37,69]. It would thus be interesting in future studies to assess whether the increase in blind-field responsiveness and coverage observed in V1 following training is associated with differences in the presence (or type) of blindsight abilities.

**pRF analysis reveals both lesion- and training-induced changes in spared V1**. Consistent with Papanikolaou and colleagues[38], we found that perilesional V1 pRFs representing blind regions of the visual field prior to training were atypically large, relative to pRFs representing perceptually-intact regions of the patients' visual field. This pattern was less pronounced in extrastriate areas (V2-V4). Enlarged V1 pRF size near the blind-field border may reflect increased excitability and enhanced plastic potential in perilesional V1 cortex, consistent with the fact that training-induced HVF improvements in CB patients generally occur along the blind-field border[21]. Consistent with this idea, single-unit electrophysiological studies in V1-lesioned animals showed increased RF size and neuronal excitability around the lesions[70,71]. A recent case study also reported larger pRF size in a hemispherectomy patient with blindsight[51]. While cortical reorganization of the underlying circuitry is not required to explain such changes in pRF properties[46], changes in V1 pRF properties near the blind-field border could indicate differences in the effective contribution of distinct subpopulations of neurons, or an imbalance in bottom-up and top-down inputs. Larger pRF size near the blind-field border could also reflect a loss of neural resolution and/or increased neural positional disorganization[72], which may explain reports of perceptual distortions and filling-in effects along the blind-field border in CB patients[73,74]. Regardless of its origin, the presence of V1 pRFs covering the blind field shows how fMRI can complement visual-field perimetry and identify perimetrically-blind regions associated with residual V1 activity that could be harnessed through training.

Critically, training-induced HVF recovery also correlated with the amount of pre-training V1 coverage of the blind field. Moreover, training was associated with a further increase in the size of the already-large pRFs along the blind-field border, which in turn, increased V1 coverage of the blind-field. This increase in visual coverage of the blind field was principally mediated by the enlargement of V1 pRFs following training, without consistent changes in the number of pRFs covering the blind field or in their position relative to the blind-field border. As with the data detailed above (i.e., atypically-large pRFs in perilesional V1 cortex of untrained CB), this result is also consistent with single-unit electrophysiology in V1-lesioned animals showing that repetitive stimulation of unresponsive border locations causes RF enlargements[71]. It is unclear whether and how visual training affects pRF properties, both in neurotypical and patient populations. Our visual rehabilitation approach requires patients to suppress eye movements and covertly attend to specific regions just

inside their blind-field border. Several studies have reported changes in the position and size of RF[75] and pRF[76–78] when spatial attention is covertly deployed to a peripheral location. Consequently, repeated, attentional deployment near the blind-field border may have altered pRF properties and increased coverage during the extended training we administered over several months. Consistent with this hypothesis, training-induced recovery of conscious luminance detection sensitivity in CB patients may depend strongly on attentional deployment during training[22,55].

**Limited changes in extrastriate visual areas following training.** Both lesion- and training-induced changes in pRF properties were clear in V1, but less pronounced in extrastriate areas (V2–V4). In addition, the amount of post-training HVF recovery correlated with pre-training V1 coverage and pre-training V1 activity in the blind field, but not with visually-evoked activity and/or pRF coverage in extrastriate areas. This mismatch in visual coverage between V1 and extrastriate areas following training may be a reason why the increase in visual-field coverage in V1 was not associated with HVF recovery at the time of the post-training session. While this pattern of results is consistent with the fact that it is damage to V1–rather than to extrastriate areas–that causes defects in conscious luminance contrast detection in CB[3,4], why are these changes observed only in V1? In answering this question, it is important to remember that pRF properties are dynamic and reflect both stimulus and cognitive factors. Differences in pRF estimates can be more pronounced in specific visual areas depending on the stimulus properties and tasks used[77,78]. For instance, a recent study[78] showed that reduced visual crowding effects following perceptual training in healthy human adults are associated with changes in pRF size in early visual cortex (V2), without clear changes in other visual areas. Furthermore, increases in pRF size with attention can be observed in either lower (e.g., V1-V3) or higher (e.g., V4, hMT+, IOG) visual areas depending on the features of the visual stimuli and task used[77,78]. Finally, we must note that our ability to assess pRF dynamics in extrastriate cortex might have been limited by the stimulus configuration used in the present study. Specifically, the lack of sustained, mean luminance blank periods may have affected the estimation of baseline activity and differences in pRF estimates, particularly given the larger pRFs and coarser retinotopic maps in extrastriate areas[79]. Given these considerations, we posit that our observed, V1-centric changes were at least partially due to our focus on HVF perimetry-defined improvements resulting from specific, localized discrimination training in the blind field, and partially due to the constraints of the retinotopic fMRI methodology employed.

**How does visual discrimination training induce HVF recovery?** Our results primarily provide insights into the neural mechanisms by which visual discrimination training reduces the area of HVF defects in chronic CB. Improvements in trained discrimination abilities were also observed post-training, but their properties were quite distinct from perimetry improvements. Among other things, recovery of visual discrimination in chronic CB is highly specific to the trained locations[19–25], a hallmark of perceptual learning[67,80,81]. In contrast, changes in HVF sensitivity occured at both trained and untrained blind-field locations, extending over much larger areas than those trained, but remaining restricted to the blind-field border, while discrimination training occurred deeper into the blind-field[20,21]. The spatial mismatch between HVF sensitivity and visual discrimination performance has been reported previously and is also observed at baseline, prior to the onset of any training[21]. Reasons behind this mismatch are likely related to differences in the characteristics of the stimuli and tasks used to measure perimetry versus visual discrimination thresholds. One possibility is that repeated, sustained deployment of attention to single, blind-field locations during discrimination training can recruit and re-engage spared, perilesional cortex mediating luminance detection sensitivity, through a combination of horizontal connections and higher-level feedback projections[82,83]. Consistent with a key role of covert spatial attention, larger training-induced recovery of luminance detection sensitivity in CB patients has been associated with stronger, functional connectivity between the occipital pole and the precuneus[55]–a region of the superior parietal lobule involved in the control of overt and covert attention in space[84].

Importantly, a fundamental role of spared V1 cortex in visual rehabilitation does not rule out a contribution of extrastriate cortex to this process. Our results show that a certain level of spared V1 activity is necessary, but not sufficient, for training to induce HVF recovery in chronic CB patients. This finding suggests that visual recovery likely involves feedback from extrastriate cortex, as typically observed in intact neural systems during training[80]. For instance, the effectiveness of visual training in inducing recovery of luminance detection sensitivity in CB is related to attentional feedback from the precuneus[55], and manipulating attention during training can potentiate recovery of visual discrimination in chronic CB fields[22]. Moreover, whereas HVF recovery might rely strongly on V1, trained CB patients can recover relatively complex visual functions[4,24,25], which presumably require processing by higher-level areas. For instance, recovery of motion direction discrimination may rely primarily on extrastriate motion-selective areas (e.g., hMT+)[85]. Consistent with this idea, a recent study showed that training-induced recovery in CB results in increased hMT+ activity to moving stimuli at trained, but not untrained, blind-field locations[19]. Interestingly, they also found that stronger pre-training hMT+ activity was associated with greater visual recovery at trained locations. Baseline (pre-training) hMT+ responses representing regions of the blind field in V1-damaged patients have been shown to originate from spared perilesional V1 cortex, as well as V1-bypassing pathways[86]. A key question for future studies is to determine why the presence of marked pre-training activity in V1 and/or hMT+ is not sufficient to mediate conscious vision before training. For instance, it would be interesting to test whether training-induced recovery of conscious visual perception relies on changes in dorsolateral prefrontal cortex, whose activity reflects conscious perceptual reports[87].

Given the dramatic impact of V1 damage on most, if not all, visual functions, and the diversity of stimuli whose processing can be restored in CB[4,7,8,12], the road to greater recovery likely depends on recruitment of and interactions between multiple visual areas, and between a diversity of recovery mechanisms[7–9,11,19,29,30]. A key role of spared, V1 cortex representing the blind field in training-induced visual recovery is not incompatible with the involvement of intact extrageniculostriate pathways, particularly those mediating blindsight abilities[7–9,11,19,29,30]. The fact that residual visual abilities in CB patients can be enhanced through rehabilitation[16,19,27,37,69] offers a vital pathway for recovery. Yet, it should be noted that whereas rehabilitation can restore some visual functions, recovered vision in CB fields remains partial and does not attain the level of perception seen in intact fields. Plasticity after cortical stroke appears to involve potentiating responses of spared neural circuits with training, rather than functional remapping of new circuits. Future rehabilitation approaches should rely on more personalized approaches that take advantage of the diversity of recovery mechanisms potentially available to each individual CB patient following cortical damage.

In conclusion, the present study reveals a hitherto unrecognized, potentially vital role of perilesional V1 cortex in mediating

training-induced recovery of conscious luminance detection sensitivity in chronic, stroke-induced CB. Our findings indicate that recovery relies on pre-training, spared, V1 activity representing blind-field locations. Moreover, training increased V1 coverage over blind-field regions, which was mediated by the enlargement of V1 pRFs in close proximity to the initial blind-field border and suggests increased responses deeper within the blind field. Aside from improving our scientific understanding of the neural mechanisms underlying visual recovery in CB, our findings are of vital importance for the development of more principled, customized, clinical neurorehabilitation therapies, adapted to the increasing number of people who suffer from cortically-induced visual impairments. Targeting blind-field regions with spared V1 activity might substantially improve the benefits and efficacy of visual restoration training, particularly when administered in the subacute period (i.e., first 3 months post-stroke) during which our training paradigm can substantially enhance the capacity for recovery[20].

## Methods

**Participants.** Eleven adult CB subjects (5 females; see Table 1 for demographics), ranging in age from 29 to 77 years old (mean: 61 ± 13 years old) underwent visual retraining starting at least 5 months (35 ± 68 months, ranging from 5 to 237 months) after stroke-induced occipital damage (verified from structural MRI; Fig. 2). These patients had large, contralesional, homonymous visual field defects determined from 24-2 and 10-2 Humphrey Visual Fields (HVFs) and shown in Fig. 2 and Supplementary Fig. S1. All patients were recruited with the intent to train, and were thus treated similarly in terms of training and testing. Recruited subjects had no ocular health problems, neurological and/or cognitive impairments, and none of the subjects suffered from visual or other forms of neglect. All 11 CB participants were scanned pre-training, but only 8 could complete the post-training fMRI session (CB1-8). CB10 had become MRI-incompatible at the time of the post-training visit, and technical difficulties with the magnet meant that post-training MRI could not be collected at the time of CB9 and CB11's post-training visits. Nine visually- and neurologically-intact subjects (7 females, mean age: 50 ± 14 years, range: 28–65 years) served as age-matched controls. The University of Rochester's Research Subjects Review Board approved all experiments conducted as part of this study; they did not designate this study as a clinical trial. The experimental procedures adhered to the tenets of the Declaration of Helsinki and were conducted after obtaining the subjects' informed, written consent. The authors affirm that human research participants provided informed consent for publication of the data shown in this study.

**Visual discrimination testing and training.** Patients were trained and tested on coarse, static orientation discrimination and/or motion direction discrimination tasks at specific locations of their visual field (Table 1 and Fig. 1; see also Supplementary Table S1). For the coarse, static orientation discrimination task, we measured percentage correct performance for discriminating the orientation of high-contrast (95% Michelson contrast), static Gabor stimuli (1 cycle/deg) that were either vertical or horizontal. Gabor stimuli were presented at a specific visual-field location, within a Gaussian aperture extending 5 deg in diameter ($\sigma = 1$ deg), and for 500 ms with slow (250 ms) onsets and offsets. For the coarse motion direction discrimination task, subjects were asked to report if random dot stimuli were moving either leftward or rightward. Random dot stimuli contained dark dots (0.5 cd/m$^2$, 0.06 deg diameter) presented within a circular aperture (5 deg diameter), moving at a speed of 5 or 10 deg/s (lifetime: 250 ms) with a density of 2.6 dots/deg$^2$. Each dot was displaced in a direction chosen randomly from a uniform distribution around either the leftward or rightward direction axis. Task difficulty was adjusted using a 3:1 staircase, which increased direction range from 0° to 360° in 40° steps after each set of 3 consecutive, correct responses, and decreased it by one 40° step for every incorrect response[22,24,25]. We computed normalized direction range (NDR) thresholds by estimating the broadest distribution of dot directions for which subjects could discriminate the global motion direction of random dot stimuli. NDR thresholds were expressed as a percentage of the maximum direction range in which dots could move (i.e., 360°). In addition to the coarse motion direction discrimination task, some subjects (i.e., CBs 6,7,9) were also trained and tested on a fine motion direction discrimination task[22]. In this version of the tasks, subjects had to discriminate whether 100%-coherent dot stimuli (i.e., with 0° distribution range of dot directions) were moving upward or downward from the horizontal axis, regardless of the leftward/rightward motion component. Task difficulty was adjusted with a 3:1 staircase, which decreased the size of the angle between the direction of motion and the horizontal meridian from ±90° to ±1°. Normalized fine direction angle thresholds were expressed as a percentage of the maximum direction angle in which stimuli could move (i.e., 90°). Normalizing both coarse direction range and fine direction angle thresholds allowed us to combine them in Fig. 1d to depict training-induced recovery in

motion discrimination. For all tasks, stimuli were presented for 500 ms and signaled by a tone. Auditory feedback was provided after each response. During in-lab testing, several visual-field locations were measured using both visual discrimination tasks to map the blind-field border under online fixation monitoring using an infrared eye tracker (Eyelink 1000 or ISCAN RK464) interfaced with our psychophysical software to ensure gaze-contingent stimulus presentation. Note that the training and testing stimuli used in the present experiments were not designed to elicit blindsight[4,24]. Thus, we can only speculate as to the presence and type of blindsight abilities in our patients. For further details about visual discrimination training and testing in our CB patients, see our published studies[20–25].

Training started at visual-field locations where discrimination performance on either task first dropped to chance levels (50% correct) during blind-field border mapping. This was a criterion for selecting specific training locations, near the blind-field border. Patients then trained at home, performing 300 trials per day, per training location, at least five days per week. Most patients started training at two non-overlapping locations of their blind field, on coarse orientation discrimination and/or on motion direction discrimination while CB6 began training at a single location (see Table 1 and Supplementary Fig. S1). Progress was assessed via weekly analysis of data log files automatically generated by our training software and sent to the lab. Once performance increased to a level comparable to that at equivalent, intact-field locations (measured during pre-testing), the training location was moved 1° deeper into the blind field along the x-axis. While home training was performed without eye-tracking, patients were repeatedly reminded of the importance of maintaining steady fixation at the central fixation spot during training. Critically, all performance on the trained tasks was verified with online eye-tracking in lab during the post-training visit, and patients were not considered to have recovered unless in-lab performance demonstrated improvement matching that recorded during home training. Note that some aspects of training and testing could vary between studies and thus patients, as detailed in each study[21–25].

**Perimetric visual field tests.** For each in-lab testing session, four total Humphrey Visual Field (HVF) tests (24-2 and 10-2 for each eye) were performed in each patient, using a Humphrey Field Analyzer (HFA II 750), by the same ophthalmic technician, at the Flaum Eye Institute of the University of Rochester Medical Center. HVF perimetry measures monocular luminance detection thresholds (in dB) by testing light intensities in a regular grid over the central 48° (24-2 HVF) or 18° (10-2 HVF) of the visual field, with fixation controlled using built-in eye-tracking. HVF thresholds (in dB) reflect the extent to which light can be dimmed from the maximum intensity (3,174 cd/m$^2$) and still be detected relative to background (~10 cd/m$^2$) at each test location. For all patients enrolled in the present study, percentage of fixation losses, false positive and false negative responses were <20%. Given the homonymous nature of their deficits, composite binocular HVFs were generated by averaging luminance sensitivity from monocular HVFs at each location between both eyes. Then, we interpolated between test locations to obtain smooth composite HVF with 0.1 deg$^2$ resolution (see our previous study[21] for more details; see Fig. 2 and Supplementary Fig. S1 for individual maps). By averaging and interpolating 4 different HVF maps, we also reduced the possible impact of small fixational eye movements, which would have had to occur consistently four separate times for locations within the central 10 deg. Several metrics calculated by the Humphrey STATPAC software (Zeiss Humphrey Systems) can be used to define visual defects and quantify HVF changes in CB patients[21]. Here, we specifically measured the area of the HVF in which changes in sensitivity (increase or decrease) occurred. To do so, differences in HVF maps were computed by first subtracting pre- and post-training, non-interpolated HVF maps, and then interpolating HVF sensitivity change ($\Delta HVF_{prepost}$) between test locations. CB fields are characterized by an abrupt fall-off in HVF sensitivity, from typically-intact HVF levels (~30 dB) to perimetrically-blind (0 dB) levels. The size of HVF deficits (Table 1) was computed as the area of impaired HVF sensitivity (i.e., impaired binocular average HVF sensitivity below 15 dB), which was used to delimitate the border of the blind field from the initial (pre-training) HVF measurement and compute a map of the distance from the blind-field border–i.e., the depth in the blind field (in deg) of voxels representing locations of CB fields. An improvement by at least +6 dB was used as the criterion for HVF recovery, and a decrease by at least -6dB for HVF worsening, similar to our previous study[21]. Note that a region of the blind field showing significant recovery (≥+6 dB) might remain impaired post-training relative to HVF sensitivity in the intact visual field. The double test-retest variability of HVF measurements indicated by the factory standard (STATPAC, Zeiss Humphrey Systems) is 3 dB. Note that we separately measured the test-retest variability of HVF measurements at the University of Rochester Medical Center, and found it to be about 2.5 dB when our subjects performed 10-2 HVF tests (see also Supplementary Fig. e-5A in our previous study[21]).

**MRI acquisition.** MRI data were acquired on a 3 T Magnetom Trio scanner (Siemens, Erlangen, Germany) using a 32-channel head coil. Functional scans were acquired with gradient recalled echo-planar imaging to measure blood oxygen level-dependent (BOLD) changes in image intensity[88]: 21 slices oriented perpendicular to the calcarine sulcus; repetition time 1.5 s; echo time 30-ms; flip angle 75°; voxel size 3 × 3 × 3 mm; grid size 64 × 64. At the beginning of each scanning session, an in-plane T1 weighted (MPRAGE, magnetization prepared rapid gradient echo) anatomical volume was acquired in the same slices as the functional scans,

but with voxel size of 0.75 × 0.75 × 3 mm. In addition, three high-resolution, T1-weighted anatomical volumes (MPRAGE, 1x1x1 mm) were acquired. The high-resolution T1 volumes were averaged together, and the average was used to extract and computationally flatten the cortical surface using FreeSurfer (version 5.3; http://surfer.nmr.mgh.harvard.edu). In-plane, anatomical images were aligned with the average high-resolution anatomical volume by an automated, robust image registration algorithm[89]. The alignment parameters were used to project the measured fMRI responses onto the flattened cortical surfaces for visualization.

**Retinotopic mapping**. Pre- and post-training retinotopic mapping sessions were identical. Traveling wave stimuli used for retinotopic mapping (Fig. 3 and Supplementary Figs. S2, S4) were presented using the MGL toolbox (version 1.5; gru.stanford.edu/doku.php/mgl/overview). Retinotopic mapping stimuli consisted of clockwise/counterclockwise rotating wedges and expanding/contracting rings filled with 100%-contrast, black-and-white sliding checkerboard stimuli. Wedges spanned 0.5-11.5 deg in radius; i.e., a small gray circle (1 deg diameter) was in the middle of the checkerboard stimuli, within which appeared a fixation cross. All subjects performed a task at fixation during scanning to maintain the patient's gaze and attention to the central fixation (see below for details). Both the rings and the wedges had a 25% duty cycle. Each run in a session consisted of 10.5 cycles (24 s length) of the stimulus rotating or expanding/contracting (168 volumes). The first 8 volumes (0.5 cycle) were discarded prior to the phase-encoded retinotopic analysis. Each scanning session consisted of six runs of the wedge stimulus (3 clockwise and 3 counterclockwise) and four runs of the ring stimulus (2 expanding and 2 contracting). The human MT/MST complex (hMT+) was defined by measuring BOLD responses to coherent vs. incoherent motion (block alternation protocol, 18 s period). The motion stimulus consisted a 10° radius circular aperture with moving white dots presented on a black background (average density of 3 dots/deg$^2$). Epochs of coherent motion consisted of 100% coherent translational motion (7 deg/s), randomly changing in direction every 3 s (6 possible motion directions). During epochs of incoherent motion, each dot was assigned a random direction on every frame of the stimulus. Each full cycle was 18 s long (12 frames) and the stimulus was run for 11 cycles (176 volumes). The first 16 volumes (1 cycle) were discarded prior to data analysis.

**Behavioral task during MRI scanning**. All subjects performed a task at fixation to minimize fixation breaks with large eye movements and to ensure consistent behavioral and attentional state throughout the fMRI data acquisition. A two-interval forced-choice (2-IFC) luminance decrement detection task was performed with respect to the fixation cross. A cyan cross was first displayed at the beginning of each trial and was briefly dimmed during each interval. The two target intervals were separated by a 500 ms period, during which the cross' color changed back again to cyan. After a final 500 ms, the cross turned yellow signaling that the subject should report which of the two intervals contained the dimmer cyan cross by pressing one of 2 buttons. After the button press, the fixation cross changed from yellow to either green for a correct response, or to red for an incorrect response as well as if the subject did not respond. The target luminance decrement was adjusted throughout each scanning session using a 2-down/1-up staircase to maintain performance near 71%-correct and maintain attention at fixation during both pre-training (71 ± 9%) and post-training (73 ± 4%) scanning sessions.

**MRI lesion reconstruction**. Lesions were defined by hand on the high-resolution, T1-weighted, anatomical images. Lesions were easily delineated on T1 anatomical images, being largely filled with CSF and/or had relatively low image intensity. Lesion reconstruction was performed using the FreeSurfer software by filling voxels corresponding to the lesioned area with an image intensity corresponding to that of white matter. These voxels were then added to the white matter mask generated by FreeSurfer, creating a surface boundary at the approximate location in the brain where gray matter had existed prior to the lesion. This procedure minimized distortions in the cortical reconstruction, and facilitated visualizing the lesions on flattened gray matter maps (e.g. orange areas in Fig. 3 and Supplementary Fig. S4).

**Phase-encoded retinotopic analysis**. All fMRI data analysis steps were performed with custom software (mrTools, http://www.cns.nyu.edu/heegerlab) written in Matlab (The MathWorks, Natick, MA). The first half-cycle of each fMRI run was not included in subsequent analysis to allow longitudinal magnetization and hemodynamics to reach steady state. The data from each run were preprocessed using standard procedures for motion compensation[89,90] and detrending[91]. The data were then analyzed by fitting a sinusoid to the time series of each voxel, and by computing the correlation/coherence between the measured time series and the best-fitting sinusoid[92,93]. Data acquired with different stimulus directions were combined to estimate the response phase independent of the phase lag caused by the hemodynamic delay of the fMRI response. Time series data for each run were first coarsely corrected for the hemodynamic delay by shifting the time series of each voxel back three time points (4.5 s). The time series for the counterclockwise wedge and contracting rings were time-reversed and averaged with the time series data for clockwise wedges and expanding rings, respectively. This way, any residual phase lag was canceled allowing us to directly convert response phase into polar angle or eccentricity without having to estimate hemodynamic delay. The two

motion localizer runs were directly averaged together. The Fourier transforms of the resulting time series were obtained and the amplitudes and phases at the stimulus frequency were examined. Coherence was computed as the ratio between the amplitude at the stimulus frequency and the square root of the sum of squares of the amplitudes at all frequencies. Coherence, amplitude and phase values from each gray matter voxel were visualized on flat maps of the occipital cortex centered on the occipital pole (Fig. 2 and Supplementary Figs. S2 and S4). V1 boundaries, as well as boundaries between both ventral and dorsal extrastriate visual-cortical areas were identified in the flattened retinotopic maps as phase reversals in polar angle components[39,40,93] and used to draw specific regions-of-interests (ROIs) on the flat maps, before being converted to cortical depth. Similar to previous retinotopic mapping studies[39,40,93], we could identify V1, ventral (e.g., V2v, V3v, V4) and dorsal (e.g., V2d, V3d, V3A/B, V7) extrastriate cortical areas, as well as the human motion-sensitive complex (hMT+), in intact hemispheres of chronic CB patients and visually-intact controls (Supplementary Fig. S2). To improve the accuracy of retinotopic maps in the damaged hemispheres of CB patients (Fig. 2 and Supplementary Fig. S4), intact anatomical landmarks and comparisons to the intact hemispheres were used. This was particularly the case for two of the patients with full hemianopia (CB9,CB11) who showed almost complete damage to V1 with no clear V1/V2 borders. Note that these two patients only participated in the pre-training fMRI session. Moreover, omitting these two patients from the analyses did not affect our main findings. On rare occasions, some extrastriate areas were not always identifiable, due either to a lack of activity or overlap with the lesion (e.g., CB6), or to an absence of clear phase reversals between areas (e.g., CB9,CB11). Coherence, amplitude and phase values of each voxel within a ROI were used for further ROI analysis. In addition, each voxel was associated to various HVF measures based on its retinotopic position (e.g., pre-training HVF sensitivity, pre-post change in HVF sensitivity, depth in the blind field). Voxels with less than 10% coherence were excluded from further analyses.

**Population receptive field (pRF) analyses and coverage maps**. The pRF method[43,44] estimates the position and size of each voxel's RF by determining the area of the visual field that best explains the relation between the time-course of the retinotopic mapping stimulus and the measured BOLD time series. The pRF model implements an isotropic Gaussian spatial receptive field, whose center and radius are derived by fitting the voxel's BOLD signal responses to the model's estimated responses elicited by convolving the model with the retinotopic mapping stimuli. pRF depict the area of the visual field that effectively evokes a response for a given voxel. pRF centers were limited to 10 deg eccentricity, and pRF size to 2/3 of the maximum eccentricity (i.e., 6.67 deg half-width radius). Similar results (e.g., significant increase pRF size and coverage of the blind field in V1 following training, limited changes in extrastriate areas) were observed when omitting these limits on the maximum pRF eccentricity and size. Voxels for which the pRF model explained less than 10% of the variance were excluded from further analyses. Using this method, we derived reliable retinotopic and pRF size maps, which allowed us to estimate coverage maps of the visual field from V1 voxels (Fig. 6). Each voxel covers a specific region of the visual field, and many points in the visual field are covered by at least one pRF. First, we combined the pRF center and size estimates from each voxel using 2D Gaussians with peak amplitude normalized to 1. pRF covering regions of the blind field corresponded to pRFs with individual normalized amplitude of at least 2/3 over blind-field regions; i.e., pre-training HVF sensitivity ≤ 15 dB (Fig. 6). Visual coverage maps were created by representing the highest pRF value at each point of the visual field[38,44,94]. Given that the peak value of the 2D Gaussian was normalized to 1, the range of values at each point of the subject's coverage map was between 0 (no coverage) and 1 (full coverage).

**Statistical analyses**. We used generalized linear mixed-effects models, with participants as a random effect, to predict the magnitude of post-training HVF recovery based on various pre-training variables (e.g., pre-training V1 response, depth in the blind field relative to the blind-field border, Fig. 4). Specifically, we selected all V1 voxels representing regions of the blind field (i.e., pre-training HVF sensitivity ≤15 dB). Each V1 voxel had a given visually-evoked response (coherence and amplitude) and represented a specific location of the blind field. That is, each voxel was associated with a blind-field location at a specific distance from the blind-field border (i.e., depth in the blind field) and with a specific change in HVF sensitivity following training. Regression models had a gamma distribution with a log-link function. Fixed-effects coefficient estimates are reported for each regression analysis along with the t-statistic and p-values, as well as with the lower and upper limits corresponding to the 95%-confidence interval for each fixed-effects coefficient of the regression models. Note that a small constant (+2.925 dB) was added to the magnitude of post-training HVF recovery to avoid non-positive values when fitting the regression model with a gamma distribution. Similar results were observed when setting non-positive values to near-zero values or when using regression models with a normal distribution, which allows non-positive HVF change values. A similar approach was used to assess differences in visually-evoked responses before and following training (Fig. 5). Generalized linear mixed-effects models with gamma distribution and log-link function, and participants as a random effect, were used to determine whether training was associated with changes in the response of V1 voxels representing the blind field, as a function of the amount of post-training HVF change associated to each voxel. We also tested

whether training resulted in a difference in the position depth in the blind field of V1 voxels, as a function of the post-training HVF change observed. Note that V1 voxels were binned for illustration purposes only, using a median split (e.g., Fig. 4f–h) or as a function of the amount of HVF change (e.g., Fig. 5b–d). A similar pattern of results was observed when using a more conservative criterion to select V1 voxels representing impaired HVF locations (e.g., using ≤6 dB instead of ≤15 dB). Standard parametric tests (i.e., repeated-measures ANOVAs, paired t-tests) were used to assess reliable within-subject differences in pRF estimates computed for each participant as a function of training (pre vs post) and HVF coverage (i.e., whether pRF covered regions of the initial blind field or solely intact-field regions, Fig. 7 and Supplementary Fig. S5). In all cases in which the Mauchly's test of sphericity indicated a violation of the sphericity assumption, Greenhouse-Geisser corrected values were used. Partial eta-squared ($\eta_p^2$) and Cohen's $d$ values were calculated to assess effect size for ANOVAs and paired-sample t-tests, respectively. Wilcoxon signed-rank tests were used when the normality assumption was violated, with the effect size reported as the rank-biserial correlation (r). In addition, a linear mixed-effects model (normal distribution), with participants as a random effect, was used to capture the linear changes in pRF size with eccentricity, as a function of training (pre. vs post) and HVF coverage (blind- vs intact-field coverage) (Fig. 7c and Supplementary Fig. S8). To illustrate the differences in pRF size as a function of pRF eccentricity, we computed mean pRF size for each participant at 4 different eccentricities (2, 4, 6 and 8 deg) using ±1 deg bins, separately for pre- and post-training and for pRFs covering the blind field or not (Fig. 7c and Supplementary Fig. S8).

**Reporting summary**. Further information on research design is available in the Nature Research Reporting Summary linked to this article.

## Data availability

All processed data reported in this paper have been deposited in a database and are publicly available as of the date of publication at www.osf.io/9n6h2. Source data are provided with this paper. Anonymised raw data and additional information are available from the corresponding author upon reasonable request. Source data are provided with this paper.

## Code availability

All software used in this study are described in the Methods. Custom code necessary to analyze the data, if any, are publicly available at www.osf.io/9n6h2. Additional information are available from the corresponding author upon reasonable request.

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

## Acknowledgements

The present study was funded by NIH (EY027314 and EY021209 to K.R.H.; Core Center Grant P30 EY001319 and training grant T32 EY007125 to the Center for Visual Science; a pre-doctoral NRSA EY025918 to MRC), an unrestricted grant from the Research to Prevent Blindness (RPB) Foundation to the Flaum Eye Institute, and a Marie Sklodowska-Curie Action Individual Fellowship to A.B. (799291). The authors thank Terrance Schaefer, who performed Humphrey visual field tests and to Patricia Weber for running the MRI on all patients presented here. We also thank Drs. Steven Feldon, Zoe Williams, Bogachan Sahin, Ania Busza, Shobha Boghani, Alexander Hartmann and Ronald Plotnick for referring CB patients to us for study.

## Author contributions

A.B., A.D., E.P.M., D.J.H., and K.R.H. designed the study; A.B., M.D.M., M.R.C., and A.D. collected the data; A.B., A.D., M.D.M., and E.P.M. analyzed the data; A.B. and K.R.H. wrote the manuscript, and all authors commented on it.

## Competing interests

KRH is co-inventor on US Patent No. 7,549,743. The remaining authors have no competing interests.
