## [Peer Review File · Nature Communications]

Reviewers' Comments:

Reviewer #1:

Remarks to the Author:

The study investigates the presence and neural determinants of training-induced recovery of luminance detection sensitivity in patients with chronic damage to V1. There are several key findings. Briefly, basic training on motion direction and orientation discrimination can induce measurable increase in luminance sensitivity, substantially reducing the extent of perimetrically blind portions of the affected visual field, especially at locations bordering the spared conscious vision. This clinical improvement is related to spared V1 activity in and around perilesional areas, and is characterized by strength of fMRI response and expansion of pRF, which increased blind field coverage.

The study is timely and extremely well conducted, with obvious implication for advancing our wisdom on the basic principles of functioning in the visual system and its plastic changes following vascular damage. This knowledge also bears direct clinical implications.

The methods are sound and the data are analyzed with state-of-the-art techniques in fMRI, and subjected to a host of different analyses that cogently focus on different aspects of the phenomenon and address various properties, thus offering a solid foundation and a coherent picture. The conclusions seem largely justified.

Overall, I have only relatively minor comments to offer.

1) The authors provide compelling evidence that increase in (conscious) visual sensitivity hinges on plastic changes and enhances responses in spared regions of the damaged V1. There is evidence, however, that recovery of visual awareness (operationalized here in terms of luminance detection sensitivity) is possible in patients with complete unilateral or even bilateral absence of V1 (e.g. Barbur et al., 1993 Brain; Giaschi et al. 2003, Silvanto 2008 Neuropsychologia). Also, V1 activity in monkeys reflects physical stimulus properties and can change independently from the conscious perception reported by the animal (Cummings et al. 1997 Nature). Likewise, a decoupling between conscious perceptual experience and variation of activity in V1 has been documented in numerous neuroimaging studies in humans. This does not detract merit at all from the present findings, but possibly requires some contextualization. The authors briefly address this issue in the intro (p.2) when referring to "blindsight" and extra-geniculo-striate pathways. I would possibly suggest to return on this in the discussion, as this may outline at least two different but non-alternative principles governing plastic changes and functional recovery after V1 damage: one that mainly exploits V1 spared tissue, and another that is more centered on V1-independent pathways.

2) related to the above point, it would be interesting to know whether patients with visual field defect were tested also for blindsight (i.e., whether they were able to guess above chance level the orientation and/or motion direction at field locations that remained clinically blind). If so, there were changes in the pre- and post-training performance at locations that remained clinically blind? There were other differences in the MRI response between cortically blind patients with and without blindsight (if tested, of course)?

3) There is little qualification on the training procedure with respect to the nature of the stimuli and tasks used for training and the recovery in luminance sensitivity. I mean, do the authors envisage a specific role for these specific tasks and stimuli, rather than for example wavelength discrimination or alike, in enhancing luminance detection? How do they explain that the task improved conscious luminance also at untrained blind-field regions (p. 4, l. 125)?

4) It would be interesting to address the role of the extra striate areas in modulating V1 spared activity. The authors mention this en passant in the discussion (p 13). Characterizing better the contribution of extra-striate areas in relation to both, the nature of training tasks and the influence on V1 changes in receptive field size, seems extant to me, though I perfectly recognize that the paper is already well focused on V1 and would be excessive to extend analyses to additional areas.

5) On p. 2 (l. 78) the Authors say that "All eleven CB patients in the present study [] had long-standing [] unilateral, homonymous, hemianopic visual-field defects". Actually, from fig 2 there were only 4 patients with full hemianopia (p. 5, 7, 8, 11). I would possibly avoid using "hemianopia" for patients that in most cases have quadrantanopia.

6) In addition to Papanikolaou 2014 PNAS (that is quote #31), I am aware of only another study

that combined retinotopic mapping with pRF measurement in a patient with cortical blindness due to hemispherectomy (Georgy et al., 2018 *Neuropsychologia*). It would be probably appropriate to quote this paper too. Also because there were some interesting analogies with some of the present results, although in a rather different context and in a patient with hemispherectomy rather than vascular damage. For example, retinotopic maps were not qualitatively altered, while there was an expansion of pRF size at eccentricities > 4 deg, and particularly in dorsal stream regions.

Minor:

- Figure 1. Label B in the figure is missing in the panel describing the two training tasks, and labels "b" and "c" in the figure should become "c" and "d" according to the legend.
- Figure 6 panel g: there are shades (I assume SEM) that seem inconsistent and the light gray line indicating "post intact" is not visible

According to Journal policy, I sign my review.
Marco Tamietto

Reviewer #3:

Remarks to the Author:

This is a very clearly and well written paper that makes a significant contribution to the field. One aspect that I think the exposition would benefit from, is a more detailed description of the stimulus sizes & training locations that were used for training the subjects. Although the basic methods are reported in Cavanaugh et al. (referred to by the authors), it would still be important to compare VF locations that received training versus VF locations that did not receive training for the subjects analyzed here. In particular, it would be interesting to show how closely trained locations correspond to the areas that show HVF recovery as well as to the areas that show the pRF/coverage changes seen by fMRI.

Stelios Smirnakis

I also have the following minor comments:

1. The test re-test variability of HVF is quoted to be about 6 dB, but it is unclear if this was tested for the subjects themselves that participated in the study or it is the general estimate from the Humphrey systems.
2. Fig 4h -- would make the grey lines slightly darker to be more easily visible.
3. the term hemianopic in lines 79, 370 is misleading as not all subjects had hemianopia (loss of vision in half the visual field).
4. Fig. 6g -- are the pre and post intact curves exactly overlapping? I could not see the post-intact curve in my printout.
5. Lines 320, 357 -- probably better to say anti-correlated to avoid confusion
6. the last sentence in lines 364-366 is confusing and would benefit from rewording. It would be good to add a graph plotting the increase in coverage following training versus the amount of training induced HVF recovery, to illustrate visually that there is no effect.

Reviewer #4:

Remarks to the Author:

This is a well-executed long-term study of the neural effects of rehabilitation training for hemianopia, by a group in a unique position to address this question. Given how little is known about this area, it is an important finding for the community and the wider fields of stroke rehabilitation and visual plasticity.

Specifically, the population receptive fields in V1 have been mapped before and after training across 8 patients with chronic hemianopia (although behavioral data and initial fMRI were acquired from 11 patients). The study is motivated by attempting to distinguish between change in V1

activity outside of the lesion and strengthening of visual pathways that avoid V1. The retinotopic mapping is very high quality with generally clear maps where it is straightforward to delineate visual areas. Furthermore, an impressive amount of data has been acquired for this study.

The study raises several issues that need a bit more explanation. The main unexpected finding is the lack of change in pRF representation in areas that show the greatest visual improvement. This seems to be counterintuitive. The results are discussed in terms of 'read-out' but surely if this is the interpretation it makes sense to also consider the pRF properties in areas V2 and V3? These areas are well-delimited in the maps and could provide additional information relating to read-out. It would be a much stronger study if comparable changes could be shown across more areas.

Conversely, there seems to be little explanation for the utility or origin of the response amplitude increase in regions that do not change behaviorally. How might this contribute to overall visual function? Why might the response amplitude be higher in regions where the patient is still blind compared to areas with recovered vision? (Figure 5).

Other comments

1. Looking at Figure 2, many of the participants have relatively small visual field loss, and in many cases, the majority of the calcarine sulcus appears to be preserved. Would it be possible include all the retinotopic maps in the main manuscript? It is, of course, more difficult to get accurate borders where a V1 lesion is large, but seeing all patients is important. Since the intact hemisphere is well defined in all patients, there are no issues with quality, so lack of clarity in the lesioned hemisphere is likely due to the damage.
2. In a similar vein, would it be possible to include in Figure 3 a participant with a large V1 lesion? CB2 appears to have very little V1 involvement, so the presence of pRFs in the blind region is perhaps not surprising.
3. Figure 4 shows the distribution of voxels relative to the blind field border. Since this is a training study, is the critical factor not the voxels around the training location? Is it possible to see the change in neural and visual performance relative to the training location(s)?
4. In terms of the larger V1 pRFs in blind regions of the visual field, is this because that region is blind due to damage in V2 or other extrastriate areas? In this case, is it that the pRF in V1 is generated by bottom-up input from LGN that would be retinotopically mapped, but receives no top-down modulation? The lack of this top-down modulation leads to an increase in pRF size, which is not helpful for perception?
5. With regard to the correlations shown in Figure 6. It is difficult to imagine that there would not be a correlation between HVF deficit and V1 coverage. Is there also a relationship between V1 lesion size and V1 coverage? How does this relate to training and post-training voxel activity? Note this is not the total lesion size, but loss of V1 tissue, which can easily be extracted by masking the lesion mask by the V1 mask. This should also be investigated in terms of the prediction of recovery potential to ensure that V1 voxel activity is not just a proxy for V1 tissue.
6. As mentioned earlier, it is a shame to only use V1 to examine pRF size and coverage. I would be very interested to see what happens in V2 and V3. Ideally I'd also see higher visual areas, but these may become challenging to define.
7. Discussion – regarding the presence of 'non-functional' retinotopic mapping of V1, see Radoeva et al (2008) Neural activity within area V1 reflects unconscious visual performance in a case of blindsight. *J. Cogn. Neurosci* 20:1927-1939. Probably should also be cited in the introduction.

Minor points

Line 48/49: a reference for rapid perceptual learning would be useful
Line 49: syllable missing in retinotopically-specific

Dear Editor and reviewers,

First of all, we would like to sincerely apologize for the unusual delay in revising our manuscript, which was due to COVID-related difficulties experienced by the first author during the past year.

We want to thank the 3 reviewers for their very thoughtful and constructive reviews of our manuscript. We have carefully addressed each of their comments. All 3 reviewers asked us to better relate our results to the training procedure (Reviewer 1 #3, Reviewer 3 #1 and Reviewer 4 #5), and asked that we expand our analyses to extrastriate areas to complement our findings in V1 (Reviewers 1 #4, Reviewer 4 #1,8). As suggested, we expanded our analyses to answer both of these points, and revised the manuscript accordingly.

Our point-by-point responses are indicated in blue below. Revisions made to the manuscript are also indicated in blue, and clearly referenced in our responses below (e.g., see lines 210-220).

We look forward to hearing back from you and hope that you will find our revised manuscript suitable for publication in *Nature Communications*.

REVIEWER COMMENTS

Reviewer #1:

The study investigates the presence and neural determinants of training-induced recovery of luminance detection sensitivity in patients with chronic damage to V1. There are several key findings. Briefly, basic training on motion direction and orientation discrimination can induce measurable increase in luminance sensitivity, substantially reducing the extent of perimetrically blind portions of the affected visual field, especially at locations bordering the spared conscious vision. This clinical improvement is related to spared V1 activity in and around perilesional areas, and is characterized by strength of fMRI response and expansion of pRF, which increased blind field coverage.

The study is timely and extremely well conducted, with obvious implication for advancing our wisdom on the basic principles of functioning in the visual system and its plastic changes following vascular damage. This knowledge also bears direct clinical implications.

The methods are sound and the data are analyzed with state-of-the-art techniques in fMRI, and subjected to a host of different analyses that cogently focus of different aspects of the phenomenon and address various properties, thus offering a solid foundation and a coherent picture. The conclusions seem largely justified.

Overall, I have only relatively minor comments to offer.

1) The authors provide compelling evidence that increase in (conscious) visual sensitivity hinges on plastic changes and enhances responses in spared regions of the damaged V1. There is evidence, however, that recovery of visual awareness (operationalized here

in terms of luminance detection sensitivity) is possible in patients with complete unilateral or even bilateral absence of V1 (e.g. Barbur et al., 1993 Brain; Giaschi et al. 2003, Silvanto 2008 Neuropsychologia). Also, V1 activity in monkeys reflects physical stimulus properties and can change independently from the conscious perception reported by the animal (Cummings et al. 1997 Nature). Likewise, a decoupling between conscious perceptual experience and variation of activity in V1 has been documented in numerous neuroimaging studies in humans. This does not detract merit at all from the present findings, but possibly requires some contextualization. The authors briefly address this issue in the intro (p.2) when referring to "blindsight" and extra-geniculo-striate pathways. I would possibly suggest to return on this in the discussion, as this may outline at least two different but non-alternative principles governing plastic changes and functional recovery after V1 damage: one that mainly exploits V1 spared tissue, and another that is more centered on V1-independent pathways.

– We concur and now provide a more complete discussion of the diversity of brain mechanisms that could underlie visual recovery following V1 damage [see lines 68; 630-642], putting our results into a broader perspective. We now also further discuss the role of V1 in visual awareness [see lines 471-479], as suggested.

2) related to the above point, it would be interesting to know whether patients with visual field defect were tested also for blindsight (i.e., whether they were able to guess above chance level the orientation and/or motion direction at field locations that remained clinically blind). If so, there were changes in the pre- and post-training performance at locations that remained clinically blind? There were other differences in the MRI response between cortically blind patients with and without blindsight (if tested, of course)?

– We now indicate that we did not systematically test for blindsight [see line 87], nor were patients trained with stimuli that would elicit blindsight [lines 104,118]. We clarified this point in the Methods [see lines 714-716] and added a brief comment on this point in the Discussion [see lines 523-532].

Note that while we can only speculate as to the presence and type of blindsight in our patients, we know, based on our published data, that chronic CB patients exhibited chance performance for all discriminations tested in the blind field prior to the onset of training. In fact, this was a criterion for selecting each training location in the blind field. A key observation that emerges from our approach is that our fMRI and psychophysical results were quite consistent across patients, suggesting that differences in blindsight abilities that may have existed prior to training had little impact on our findings. Finally, we now discuss the possibility that some of our results could be associated with changes in blindsight abilities [see lines 523-525].

3) There is little qualification on the training procedure with respect to the nature of the stimuli and tasks used for training and the recovery in luminance sensitivity. I mean, do the authors envisage a specific role for these specific tasks and stimuli, rather than for example wavelength discrimination or alike, in enhancing luminance detection? How do they explain that the task improved conscious luminance also at untrained blind-field regions (p. 4, l. 125)?

– These are important points mentioned by all reviewers (see also Reviewer 2 #1 and Reviewer 4 #5). We now provide more detail regarding the training procedures [see

Methods–Visual discrimination testing and training], but a detailed analysis of the relative efficacy of different training tasks and stimuli (particular in the color domain) for eliciting changes in luminance detection is not possible within the scope of the present paper. However, we have expanded the Discussion to further describe the results of Cavanaugh & Huxlin (2017) and the consistently-observed spatial mismatch between HVF recovery and visual discrimination improvements seen presently **[see lines 593-600]**. Note that a consistent spatial mismatch between luminance detection (i.e. perimetry) and psychophysically-measured visual discrimination performance is also frequently observed at baseline, prior to the onset of visual training (Cavanaugh and Huxlin, 2017). The reasons behind this mismatch are currently a matter of speculation. As we now discuss **[see lines 600-609]**, possible causes include key differences in stimuli and tasks used to measure perimetry *versus* visual discrimination performance, as well as a key differences in attentional deployment to the patient's blind field during training as opposed to perimetry.

4) It would be interesting to address the role of the extra striate areas in modulating V1 spared activity. The authors mention this en passant in the discussion (p 13). Characterizing better the contribution of extra-striate areas in relation to both, the nature of training tasks and the influence on V1 changes in receptive field size, seems extant to me, though I perfectly recognize that the paper is already well focused on V1 and would be excessive to extend analyses to additional areas.

– We agree that this is an interesting question (see also Reviewer 4 #1,8) and we now report fMRI results for extrastriate areas V2, V3, V4 **[see lines 391-413, Fig.8 and Figs.S7, S8]**. We show that the changes in pRF size and in blind-field coverage, as well as the correlation between pre-training blind-field coverage and post-training improved HVF area, were strongest in V1, and less clear in extrastriate areas. We revised the discussion **[see lines 568-590]** regarding the contribution of extrastriate areas and the apparent key role of V1 in training-induced HVF recovery.

5) On p. 2 (l. 78) the Authors say that "All eleven CB patients in the present study [] had long-standing [] unilateral, homonymous, hemianopic visual-field defects". Actually, from fig 2 there were only 4 patients with full hemianopia (p. 5, 7, 8, 11). I would possibly avoid using "hemianopia" for patients that in most cases have quadrantanopia.

–We thank the reviewer for pointing this out and we now characterize patients as having full hemianopia (CB7,9,11), partial hemianopia (CB1,3,4,5,8,10) or quadrantanopia (CB2,6) **[see lines 85-87]**. To avoid any confusion, we now use the term cortically-blind (or CB) rather than hemianopic throughout the manuscript.

6) In addition to Papanikolaou 2014 PNAS (that is quote #31), I am aware of only another study that combined retinotopic mapping with pRF measurement in a patient with cortical blindness due to hemispherectomy (Georgy et al., 2018 Neuropsychologia). It would be probably appropriate to quote this paper too. Also because there were some interesting analogies with some of the present results, although in a rather different context and in a patient with hemispherectomy rather than vascular damage. For example, retinotopic maps were not qualitatively altered, while there was an expansion of pRF size at eccentricities > 4 deg, and particularly in dorsal stream regions.

– This is indeed a relevant study, which we have now added to the manuscript [see lines 541-542]. In addition, we have expanded the discussion of the differences in pRF properties we observed [see lines 542-550].

Minor:

- Figure 1. Label B in the figure is missing in the panel describing the two training tasks, and labels "b" and "c" in the figure should become "c" and "d" according to the legend.
– fixed

- Figure 6 panel g: there are shades (I assume SEM) that seem inconsistent and the light gray line indicating "post intact" is not visible
– we fixed this Figure (Fig.7c; and Fig.S8) to make sure all conditions are clearly visible and differentiable. We also checked that all information is provided in the Figure legend (e.g., ± 1 SEM).

Reviewer #2:

This is a very clearly and well written paper that makes a significant contribution to the field.

1-One aspect that I think the exposition would benefit from, is a more detailed description of the stimulus sizes & training locations that were used for training the subjects. Although the basic methods are reported in Cavanaugh et al. (referred to by the authors), it would still be important to compare VF locations that received training versus VF locations that did not receive training for the subjects analyzed here. In particular, it would be interesting to show how closely trained locations correspond to the areas that show HVF recovery as well as to the areas that show the pRF/coverage changes seen by fMRI.

- Please see our response to Reviewer 1 #3, since this important point was raised by him also (and by Reviewer 4 #5). These details are now provided [see **Methods–Visual discrimination testing and training**].

With respect to the second point, although we would like to be able to perform the analysis suggested by Reviewer 2 over trained locations, this is not uniformly possible because for about half the patients, our relatively large training stimuli and multiple training locations fell at least partially if not totally outside the fMRI scanning area. Training stimuli were indeed generally located deeper into the blind field than the perimetrically-defined border of the deficit. As such, the training locations – as a group - covered too few voxels for a reliable analysis, with many partially or totally outside the field of view of the analysis (now shown in Fig.S1). Visual perimetric improvements derived from HVF were [uniformly] amenable for correlation with retinotopic analysis performed here, since they started at the border of the deficit and could thus be more completely captured by the limited visual field aperture in the magnet. As mentioned in response to Reviewer 1 #3, we now further describe the results of Cavanaugh & Huxlin (2017) and have clarified the spatial mismatch between HVF recovery and the locations of visual discrimination training and improvements [see lines 593-600].

I also have the following minor comments:

2. The test re-test variability of HVF is quoted to be about 6 dB, but it is unclear if this was tested for the subjects themselves that participated in the study or it is the general estimate from the Humphrey systems.

– We now clarify that the test-retest variability of the HVF measurements corresponds to the factory standard [see line 762] but also indicate that we measured this test-retest variability ourselves and found it to be about 2.5dB for our CB subjects when performing 10-2 HVF tests [see line 763-765].

3. Fig 4h -- would make the grey lines slightly darker to be more easily visible.

– fixed

4. the term hemianopic in lines 79, 370 is misleading as not all subjects had hemianopia (loss of vision in half the visual field).

–We agree and have fixed this throughout the text, as indicated in response to Reviewer 1 #5 [see lines 85-87].

5. Fig. 6g -- are the pre and post intact curves exactly overlapping? I could not see the post-intact curve in my printout.

–We revised this figure to help differentiate the pre and post intact curves, which are indeed overlapping relative to pRFs covering the blind field (now Fig.7c).

6. Lines 320, 357 -- probably better to say anti-correlated to avoid confusion

– we agree and now use anti-correlated when appropriate (e.g., line 368).

7. the last sentence in lines 364-366 is confusing and would benefit from rewording. It would be good to add a graph plotting the increase in coverage following training versus the amount of training induced HVF recovery, to illustrate visually that there is no effect.

– We developed this part (see lines 370-377) and added the suggested graph (Fig.7i).

Reviewer #4:

This is a well-executed long-term study of the neural effects of rehabilitation training for hemianopia, by a group in a unique position to address this question. Given how little is known about this area, it is an important finding for the community and the wider fields of stroke rehabilitation and visual plasticity. Specifically, the population receptive fields in V1 have been mapped before and after training across 8 patients with chronic hemianopia (although behavioral data and initial fMRI were acquired from 11 patients). The study is motivated by attempting to distinguish between change in V1 activity outside of the lesion and strengthening of visual pathways that avoid V1. The retinotopic mapping is very high quality with generally clear maps where it is straightforward to delineate visual areas. Furthermore, an impressive amount of data has been acquired for this study.

1. The study raises several issues that need a bit more explanation. The main unexpected

finding is the lack of change in pRF representation in areas that show the greatest visual improvement. This seems to be counterintuitive. The results are discussed in terms of 'read-out' but surely if this is the interpretation it makes sense to also consider the pRF properties in areas V2 and V3? These areas are well-delimited in the maps and could provide additional information relating to read-out. It would be a much stronger study if comparable changes could be shown across more areas.

– We agree that this is an interesting question (see also Reviewer 1 #4) and now report results for extrastriate areas V2,V3,V4 [see lines 391-413, Fig.8 and Figs.S7,S8], which revealed that the changes in pRF size and blind-field coverage, as well as the correlation between pre-training blind-field coverage and post-training improved HVF area, were primarily observed in V1, and were less clear in extrastriate areas. We revised the discussion [see lines 568-590] regarding the contribution of extrastriate areas and the apparent key role of V1 in training-induced HVF recovery. We have also expanded the discussion regarding the changes in pRF size and V1 coverage, relative to the pattern of HVF recovery observed [see lines 523-532, 542-550].

2. Conversely, there seems to be little explanation for the utility or origin of the response amplitude increase in regions that do not change behaviorally. How might this contribute to overall visual function? Why might the response amplitude be higher in regions where the patient is still blind compared to areas with recovered vision? (Fig 5).

– We have revised the discussion to explicitly mention that the origin and functional implications of these changes are unclear [see lines 516-517], although we do provide some speculation as to their potential roles [see lines 518-523]. Specifically, we hypothesize that changes in response amplitude (and pRF coverage) observed at blind-field locations that are still “blind” on post-training HVF could serve as a new “baseline” that could be recruited with subsequent training, and lead to further recovery. This would be consistent with behavioral findings showing recovery deeper within the blind field with additional training. Moreover, as suggested by Reviewer 1 (#2), we now also consider the potential role of blindsight abilities and the fact that these changes – while not associated with recovery of conscious luminance detection sensitivity *per se* – may reflect changes in residual unconscious abilities (which remain to be assessed) [see lines 523-532].

Other comments:

3. Looking at Fig 2, many of the participants have relatively small visual field loss, and in many cases, the majority of the calcarine sulcus appears to be preserved. Would it be possible include all the retinotopic maps in the main manuscript? It is, of course, more difficult to get accurate borders where a V1 lesion is large, but seeing all patients is important. Since the intact hemisphere is well defined in all patients, there are no issues with quality, so lack of clarity in the lesioned hemisphere is likely due to the damage.

– As suggested, we now include our original Figure S2 as a main figure (**new Figure 3**) to show all the retinotopic maps for all the damaged hemispheres prior to training. We should point out that while we only show visual field defects over the area of the visual field stimulated during fMRI (by necessity, relatively small), the visual field loss in our group of participants was not small and only two patients (CB2, CB6) had quadrantanopias (see **Fig.S1**). The majority had either partial or complete hemianopias. We clarified this point in the manuscript [see lines 85-87], as noted in our responses to other Reviewers (R1#5, R2#4).

4. In a similar vein, would it be possible to include in Figure 3 a participant with a large V1 lesion? CB2 appears to have very little V1 involvement, so the presence of pRFs in the blind region is perhaps not surprising.

– We agree and have replaced the original figure with a **new Figure 3** that shows all the retinotopic maps analyzed in the present study. In addition, we now show pRF plots in our **new Figure 6**, with examples from 3 patients (CB1, CB4 and CB6) who have very clear damage to V1.

5. Figure 4 shows the distribution of voxels relative to the blind field border. Since this is a training study, is the critical factor not the voxels around the training location? Is it possible to see the change in neural and visual performance relative to the training location(s)?

– This important point was also raised by Reviewers 1 (#3) and 2 (#1). As indicated in our earlier responses to these reviewers, there is a spatial mismatch between visual field locations trained on visual discriminations and the pattern of training-induced HVF recovery. This was reported in detail previously (Cavanaugh & Huxlin, 2017). Nonetheless, we ascertained and now indicate that the distance from training locations had no effect on the pattern of HVF recovery [see lines 229-232; Fig.S3f]. We also provide more details in the Methods regarding our training stimuli and procedure [see **Methods–Visual discrimination testing and training**], and the fact that one reason we could not perform the analysis suggested is that for many participants, the training stimuli fell outside the field of view of the magnet during imaging, now shown in **Fig.S1** [see lines 231-232]. In contrast, HVF improvements always fell within the field of view of the magnet. Nonetheless, we acknowledge the fact that a better way to answer this question [than retinotopic mapping] may be for future studies to measure and compare changes in fMRI activity at specific trained and untrained locations *while* patients actually perform the visual discrimination task in the magnet [see lines 496-500]. Finally, we added a brief paragraph to mention a study released a few days ago in The Journal of Neuroscience (Ajina et al., *in press*), showing that training-induced recovery in CB patients results in increased hMT+ activity at the trained but not untrained locations [see lines 620-627].

6. In terms of the larger V1 pRFs in blind regions of the visual field, is this because that region is blind due to damage in V2 or other extrastriate areas? In this case, is it that the pRF in V1 is generated by bottom-up input from LGN that would be retinotopically mapped, but receives no top-down modulation? The lack of this top-down modulation leads to an increase in pRF size, which is not helpful for perception?

– These are interesting questions. We have expanded the Discussion to provide further details regarding the presence of larger pRFs along the blind-field border prior to training [see lines 542-550]. Specifically, we clarified that pRF dynamics do not necessarily reflect reorganization of the underlying circuitry, and now discuss possible bottom-up and top-down factors that could mediate enlargements near the blind-field border. Related to this point, we have also expanded the Discussion regarding the further pRF enlargement observed in V1 following training [see lines 551-557, 565-567, 576-584]. The fact that visual recovery is strongly related to attentional deployment during training (something shown to increase pRF size in neurotypical observers) suggests an important contribution of top-down modulation in this phenomenon.

7. With regard to the correlations shown in Figure 6. It is difficult to imagine that there would not be a correlation between HVF deficit and V1 coverage. Is there also a relationship between V1 lesion size and V1 coverage? How does this relate to training and post-training voxel activity? Note this is not the total lesion size, but loss of V1 tissue, which can easily be extracted by masking the lesion mask by the V1 mask. This should also be investigated in terms of the prediction of recovery potential to ensure that V1 voxel activity is not just a proxy for V1 tissue.

– As suggested, we estimated the loss of V1 tissue by computing the percent loss of V1 volume between the intact and lesioned hemispheres of each patient. As expected, a stronger loss of V1 tissue was associated with a larger HVF deficit area as well as with smaller V1 coverage of the blind field (see **Fig.7**). However, the loss of V1 tissue did not correlate with the amount of post-training HVF recovery. Moreover, the amount of HVF recovery was not correlated with the number of voxels located within the blind field, or with the number of pRFs covering blind-field areas prior to training. These findings supports the notion that HVF deficits are caused by V1 damage, but that visual recovery following training does not strictly depend on the size of the deficit or on the loss of V1 volume [**see lines 375-382**].

8. As mentioned earlier, it is a shame to only use V1 to examine pRF size and coverage. I would be very interested to see what happens in V2 and V3. Ideally I'd also see higher visual areas, but these may become challenging to define.

–We agree and now provide additional data for extrastriate areas V2, V3 and V4. Interestingly, the changes in pRF size and coverage of the blind field were pronounced in V1, but less clear in extrastriate areas. Although these findings still support a tighter link between V1 and luminance detection sensitivity, we do mention some of the limitations of our experimental design regarding the estimation of pRF size in higher-level visual areas [**see lines 584-590**]. Comparing BOLD responses to visual stimuli presented at a specific locations of the blind field before and following training might be more adapted to higher visual areas given the larger receptive fields and coarser retinotopic maps in extrastriate cortex. **lines 408-413**].

9. Discussion – regarding the presence of ‘non-functional’ retinotopic mapping of V1, see Radoeva et al (2008) Neural activity within area V1 reflects unconscious visual performance in a case of blindsight. J. Cogn. Neurosci 20:1927-1939. Probably should also be cited in the introduction.

– Very relevant paper. We added this reference in several locations, where appropriate [e.g., lines 59,177,296,451,462].

Minor points:

Line 48/49: a reference for rapid perceptual learning would be useful

– We added references for rapid perceptual learning in intact visual systems [line 50]

Line 49: syllable missing in retinotopically-specific

– fixed [line 50]

Reviewers' Comments:

Reviewer #1:

Remarks to the Author:

The revised version has taken into account all my previous comments carefully. The study is timely, extremely interesting and well conducted. I have no further comments to offer and I congratulate the authors for a study that looks to me ready for publication.

Reviewer #3:

Remarks to the Author:

This is a timely and very well performed study that makes a significant contribution in our understanding of how responses in area V1 and extrastriate areas recover with training in subjects with dense visual field defects following cortical lesions. The authors responded in detail and appropriately to the reviewer comments within the context of the scope of the experiment they performed. When the data could not answer directly the questions asked, as in definitively identifying the mechanisms for the observed spatial mismatch between HVF sensitivity and visual discrimination performance, thoughtful discussion was added pointing out new directions likely to be fruitful paths for future research.